# An NAD⁺-dependent novel transcription factor controls stage conversion in *Entamoeba*

Dipak Manna[1], Christian Stephan Lentz[2†], Gretchen Marie Ehrenkaufer[1†], Susmitha Suresh[1], Amrita Bhat[1], Upinder Singh[1,3]*

[1]Division of Infectious Diseases, Department of Internal Medicine, Stanford University School of Medicine, Stanford, United States; [2]Department of Pathology, Stanford University School of Medicine, Stanford, United States; [3]Department of Microbiology and Immunology, Stanford University School of Medicine, Stanford, United States

**Abstract** Developmental switching between life-cycle stages is a common feature among parasitic pathogens to facilitate disease transmission and pathogenesis. The protozoan parasite *Entamoeba* switches between invasive trophozoites and dormant cysts, but the encystation process remains poorly understood despite being central to amoebic biology. We identify a transcription factor, Encystation Regulatory Motif-Binding Protein (ERM-BP), that regulates encystation. Down-regulation of ERM-BP decreases encystation efficiency resulting in abnormal cysts with defective cyst walls. We demonstrate that direct binding of NAD⁺ to ERM-BP affects ERM-BP conformation and facilitates its binding to promoter DNA. Additionally, cellular NAD⁺ levels increase during encystation and exogenous NAD⁺ enhances encystation consistent with the role of carbon source depletion in triggering *Entamoeba* encystation. Furthermore, ERM-BP catalyzes conversion of nicotinamide to nicotinic acid, which might have second messenger effects on stage conversion. Our findings link the metabolic cofactors nicotinamide and NAD⁺ to transcriptional regulation via ERM-BP and provide the first mechanistic insights into *Entamoeba* encystation.

DOI: https://doi.org/10.7554/eLife.37912.001

*For correspondence:
usingh@stanford.edu

†These authors contributed equally to this work

Competing interests: The authors declare that no competing interests exist.

## Introduction

Modulation of gene expression plays a crucial role during stage conversion in all organisms (*Gomez et al., 2010*; *Morf et al., 2010*; *Kramer, 2012*). Developmentally regulated genes can be controlled by multiple pathways including transcriptional control, post-transcriptional modification, and RNA transport, stability, or translation efficiency (*Day and Tuite, 1998*; *Spitz and Furlong, 2012*). Although transcriptional regulatory networks have been extensively studied in model organisms, our understanding of transcriptional regulation in protozoan parasites is relatively limited. Studies in *Plasmodium* (*Cai et al., 2012*), *Toxoplasma* (*Bougdour et al., 2008*; *Joyce et al., 2013*), *Giardia* (*Einarsson et al., 2015*), *Trypanosome* (*Clayton, 2014*) and *Entamoeba* (*Ehrenkaufer et al., 2013*) hint at unique pathways that are exploited by parasitic protozoa to regulate their developmental cascades.

*Entamoeba histolytica* is an anaerobic pathogen that causes invasive disease in millions of people worldwide and estimated to kill more than 55,000 people each year (*Haque et al., 2003*; *Lozano et al., 2012*). Infection with *Entamoeba* starts with the ingestion of mature cysts with contaminated food or water; trophozoites are released from the cysts in the small intestine and migrate to the colon where they proliferate. Trophozoites invade the colonic epithelium resulting in the

clinical syndrome of dysentery or amoebic colitis (*Lozano et al., 2012*). Due to unknown stimuli, some trophozoites in the colon are triggered to initiate stage conversion and transform to cysts. The cysts are passed in feces, are resistant to environmental extremes, and are able to transmit disease (*Jones and Newton, 1950*). Thus, stage conversion is crucial to parasite biology and is necessary for both pathogen transmission and disease pathogenesis.

Developmental studies in *Entamoeba* are performed in *Entamoeba invadens*, a reptilian parasite in which encystation can be induced by transfer to nutrient-deficient media (*Thepsuparungsikul et al., 1971*; *Rengpien and Bailey, 1975*; *Vázquezdelara-Cisneros and Arroyo-Begovich, 1984*). Previously, Coppi et al demonstrated several lines of evidence that an autocrine catecholamine system is involved in *Entamoeba* encystation (*Coppi et al., 2002*). Mi-ichi et al reported that cholesteryl sulfate plays an important role in *Entamoeba* encystation (*Mi-ichi et al., 2015*). While this system has been vital to increasing our understanding of amebic biology, the key regulators that control stage conversion are still not well understood. Recent RNA-Seq and microarray data from *E. invadens* during encystation offers an opportunity to identify the molecular triggers involved in regulating stage conversion in *Entamoeba* (*De Cádiz et al., 2013*; *Ehrenkaufer et al., 2013*). In this study, we utilized this RNA-Seq data and bioinformatics approaches to identify a transcription factor that binds to CAACAAA motif in gene promoter and regulates *Entamoeba* stage conversion. The *Encystation Regulatory Motif-Binding Protein* (ERM-BP) has a nicotinamidase domain; we identified that $NAD^+$ levels increase during encystation and mediate ERM-BP binding to DNA, and that increased $NAD^+$ augments encystation efficiency. This work provides the first molecular link between a metabolic coenzyme $NAD^+$ and regulation of stage conversion in a parasitic protozoan and represents a substantial advancement in a key area of parasite biology.

## Results

### Identification of consensus promoter motifs in cyst-specific genes

In order to define the transcriptional network associated with stage conversion, we utilized RNA-Seq data of different developmental stages of *E. invadens* (*Ehrenkaufer et al., 2013*). We started with 616 genes that had low expression in trophozoites but were upregulated at 24 hr of encystation. The expression data of these transcripts during the entire developmental cascade are shown in *Figure 1—source data 1*. In order to identify whether these genes are coordinately upregulated, we used bioinformatics analysis with MEME and MAST to identify conserved promoter motifs. Enrichment of the motifs within the promoters of the cyst-specific genes relative to the number of occurrences in the entire promoter set of *E. invadens* (as determined using the hypergeometric distribution, p<0.001) was prioritized for our analysis. Our analysis identified nine motifs significantly enriched among the promoters of these 616 cyst-specific genes (*Figure 1—figure supplement 1*).

### Motif-2 specifically binds to cyst nuclear protein(s)

We screened all nine motifs by Electrophoretic Mobility Shift Assay (EMSA) using crude nuclear extracts (NE) from both trophozoites and 24 hr encysting parasites. All the oligonucleotides used in EMSA are listed in *Figure 1—source data 2*. Out of nine motifs identified, only Motif-2 (CAACAAA) (*Figure 1A*) showed strong and specific binding to protein(s) from cyst nuclear extracts (*Figure 1—figure supplement 2*). This motif, which we labeled as *Encystation Regulatory Motif* (ERM), is found in the promoters of 131 cyst-specific genes (*Figure 1—source data 3*).

In order to identify the critical residues in the ERM, three mutants were generated by changing the conserved nucleotides CA to TG (*Figure 1C*). Mut-2 was able to out-compete the labeled ERM-WT probe at 10 and 100-fold excess of cold competitor (10 and 100X) in EMSA assay. In this particular gel, Mut-2 acts as a better competitor at 10-fold excess compared to ERM-WT, but was similar to WT competitor at 100X excess. Importantly, we have noted in multiple other gel shift assays that Mut-2 acts similar to WT-motif as a competitor (data not shown). In contrast, Mut-1 and Mut-3 showed no ability to compete at 10 and 100X, suggesting that changing the C at position four and A at position five in ERM are crucial for binding to cyst nuclear protein. We also observed that the ERM-core 7-nucleotide motif (without linker) was not able to compete, suggesting that this short

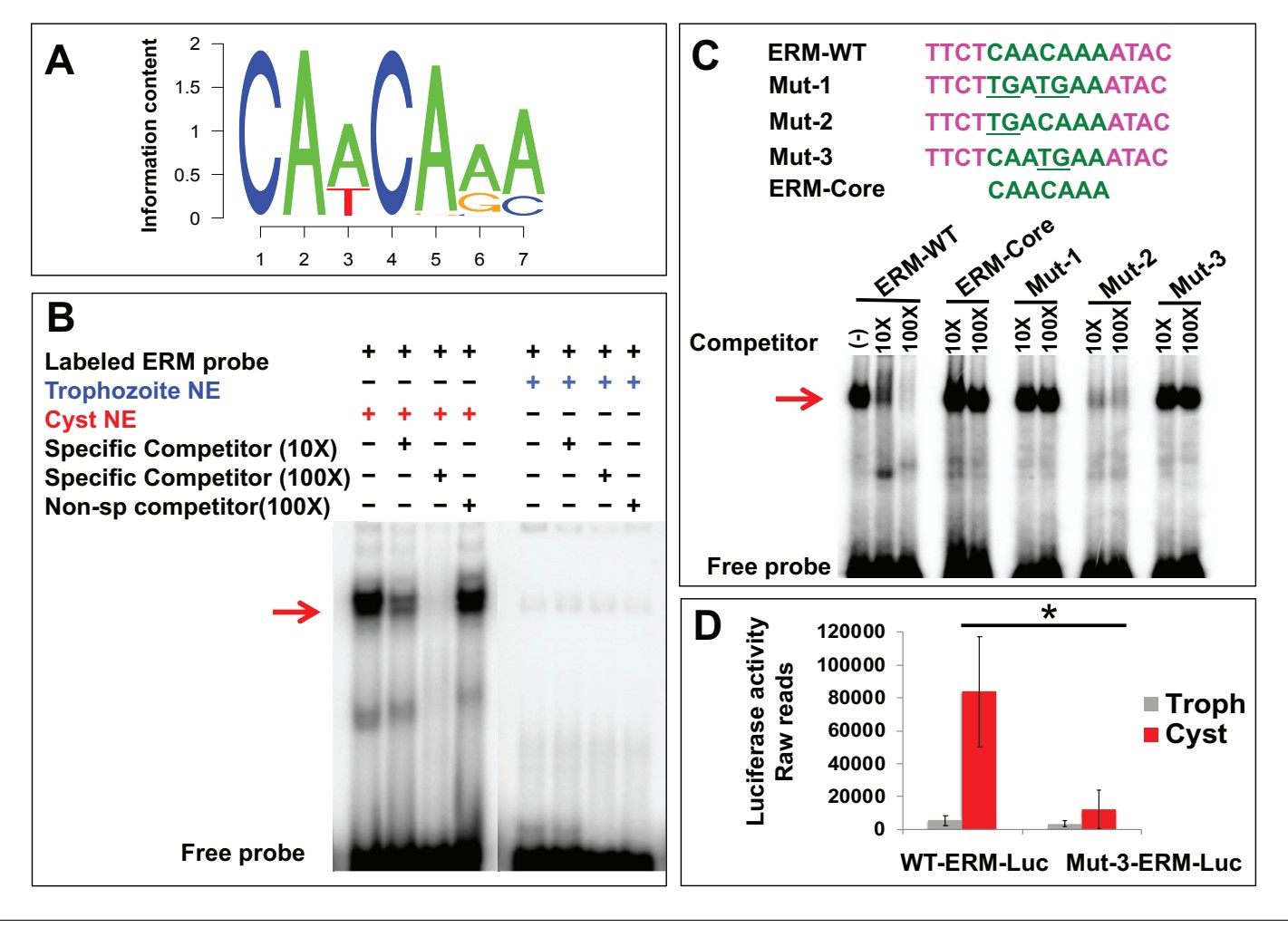

**Figure 1.** Encystation regulatory motif (ERM) specifically binds cyst nuclear protein and identification of crucial residues important for promoter activity. (A) Sequence logo of ERM, which is enriched in the promoter of cyst-specific genes. The seven-nucleotide motif information content is shown. (B) Representative EMSA results are shown in the presence and absence of different components marked as '+' and '−' respectively. Radiolabeled ERM probe was used in each reaction. Unlabeled ERM probe at 10X and 100X was used as a specific competitor and non-specific cold competitor was used at 100X as indicated. The red arrow indicates the major specific band in the gel shift assay; free probe is at the bottom. (C) Sequences of three mutants (Mut-1, 2 and 3) generated by changing the conserved CAs to TG (underlined) and ERM-core, without nonspecific flanking region are shown. Competition assays using cyst nuclear protein were performed using 10X and 100X of cold competitor against radiolabeled ERM-WT probe. The red arrow indicates the major bands that exhibit specific binding. (D) Data represents the raw luciferase readings in Trophozoites (Troph) and Cysts in WT-ERM-promoter construct and Mut-3-ERM-promoter construct having luciferase as a reporter gene. Data are mean ±s.d. (n = 3) Student's t-test; *p<0.05.

DOI: https://doi.org/10.7554/eLife.37912.002

The following source data and figure supplements are available for figure 1:

**Source data 1.** List of genes specifically upregulated during 24 hr of encystation.
DOI: https://doi.org/10.7554/eLife.37912.005

**Source data 2.** List of oligonucleotides used in electrophoretic mobility shift assay (EMSA), cloning and RT-PCR.
DOI: https://doi.org/10.7554/eLife.37912.006

**Source data 3.** List of genes having ERM (CAACAAA) in the promoter.
DOI: https://doi.org/10.7554/eLife.37912.007

**Figure supplement 1.** Motifs enriched in the promoters of genes that upregulated during 24 hr of encystation.
DOI: https://doi.org/10.7554/eLife.37912.003

**Figure supplement 2.** Summary of screening results of the motifs by electrophoretic mobility shift assay (EMSA).
DOI: https://doi.org/10.7554/eLife.37912.004

motif is not efficient at binding to nuclear protein, and flanking sequences are required for physical interaction as seen in previous studies (*Hackney et al., 2007*).

To find out whether the ERM motif is functional in the context of a full-length promoter, we used a luciferase reporter construct driven by a cyst-specific promoter that contains the WT-ERM. We have previously demonstrated that this construct increases expression during encystation (*Manna et al., 2014*). A Mut-3-ERM-Luciferase construct was generated by changing CA to TG at fourth and fifth positions. Stable cell lines were generated by transforming luciferase reporter constructs into *Entamoeba*, and luciferase assays were performed in trophozoites and cysts. We see some basal level of luciferase readings in trophozoites for both WT and Mut-3-ERM. However, mutation of the ERM in this promoter significantly decreased cyst-specific luciferase expression, suggesting that the cyst-specific activity of the full-length promoter is regulated through ERM (*Figure 1D*).

## Identification of ERM-binding protein

To identify the ERM-binding protein(s), EMSA shifted bands were excised and analyzed by LC-MS as described in the Materials and methods. The protein content of the positive shifted band (ERM-WT with cyst extract) was compared with the same gel mobilities in negative controls (ERM-core with cyst extract and ERM-WT with trophozoite extract). All proteins identified in three biologically independent LC-MS analyses are listed in *Figure 2—source data 1* and a summary of data analyses is in *Figure 2—figure supplement 1*. However, only six proteins reproducibly occurred in all three experiments which are exclusively present or enriched in ERM-WT compared to the negative controls (trophozoite extract EMSA or ERM-core EMSA) (*Figure 2—source data 1*). Out of the six proteins, four (EIN_327770, EIN_380710, EIN_083100, EIN_024000) are hypothetical proteins, one is a WD-repeat protein (EIN_359470) and one is a tyrosine phosphatase (EIN_381540). RNA-Seq data shows that hypothetical proteins, EIN_083100 and EIN_024000, are developmentally regulated with enrichment in *E. invadens* cysts (*Table 1*).

To assay binding of these six proteins to the ERM motif, we expressed GST-tagged recombinant proteins in bacteria, which were purified and used in EMSA. Out of the six recombinant proteins only recombinant protein EIN_083100 (previously named EIN_052150) bound to ERM (*Figure 2A*). EIN_083100 is annotated as a hypothetical protein and expression data (RNA-Seq and microarray) shows increased expression during encystation (*De Cádiz et al., 2013*; *Ehrenkaufer et al., 2013*). Our RT-PCR results show that EIN_083100 RNA expression is undetectable in trophozoites and upregulated at 24 hr of encystation (*Figure 2—figure supplement 2A*) whereas *Ehrenkaufer et al., 2013* showed that EIN_083100 was regulated at later time point of encystation (72 hr). Whether this difference is due to technical differences in RNA-Seq versus RT-PCR or changes in parasites that have occurred over several years is not clear; however, the EIN_083100 gene is clearly developmentally regulated. This protein is highly conserved among all other *Entamoeba* species that form cysts. Significantly, RNA expression of the *E. histolytica* homolog (EHI_146360) was upregulated in *E.*

**Table 1.** Summary of proteins identified by LC-MS and confirmation.

Six proteins were identified in all three experiments as enriched in the ERM-WT sample and shown here with Gene ID and annotations. All six proteins were expressed in bacteria and the only recombinant protein (EIN_083100) bind to ERM. Three proteins were expressed in *E. invadens* and their localization in the parasite is summarized. '*' indicates the genes which are stage specifically expressed and upregulated during encystation and 'ND' indicates 'not done'.

| No | Gene ID | Proteins | Recombinant protein bind to ERM | Overexpression in Amoeba | Localization |
|---|---|---|---|---|---|
| 1 | EIN_212680 | WD-repeat protein | No | Yes | Cytoplasm |
| 2 | EIN_192460 | hypothetical | No | ND | |
| 3 | EIN_224750 | Protein-tyrosine phosphatase | No | ND | |
| 4 | EIN_224120 | Hypothetical | No | Yes | Cytoplasm |
| *5 | EIN_083100 | Hypothetical | Yes | Yes | Nucleus/ Cytoplasm |
| *6 | EIN_024000 | Hypothetical | No | ND | |

DOI: https://doi.org/10.7554/eLife.37912.008

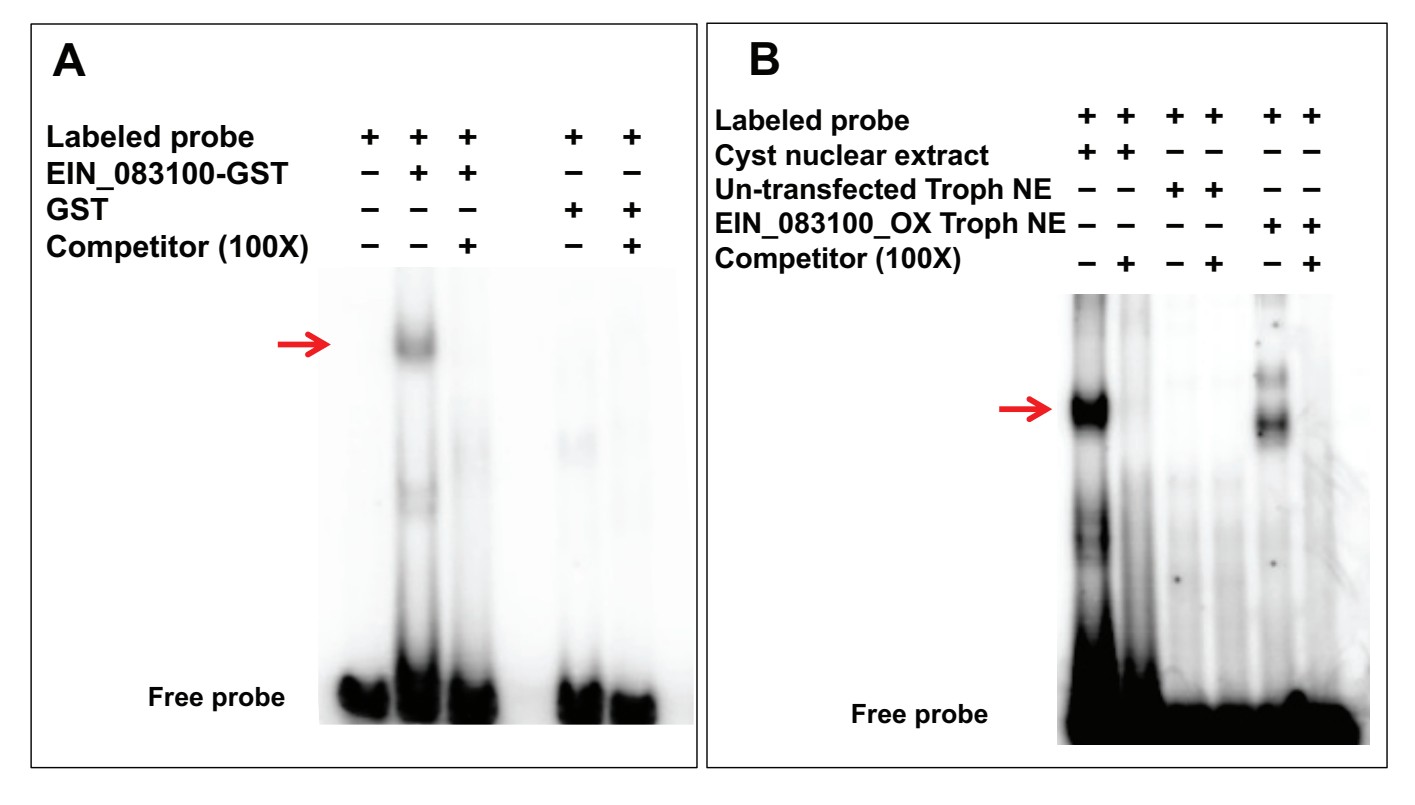

**Figure 2.** Confirmation of ERM-binding protein (ERM-BP). (**A**) EMSA results with purified GST tagged ERM-BP and radiolabeled ERM probe. Unlabeled ERM oligonucleotides in 100X excess were used as a specific competitor and GST as control protein. The red arrow indicates the major shifted band that exhibits specific binding. (**B**) EMSA results with radiolabeled ERM and nuclear extracts from both cysts/trophozoites from control cells and nuclear extracts from myc-tagged ERM-BP_OX trophozoites. Unlabeled ERM oligonucleotide in excess at 100X was used as a specific competitor. The red arrow indicates the major band that exhibits specific binding.

DOI: https://doi.org/10.7554/eLife.37912.009

The following source data and figure supplements are available for figure 2:

**Source data 1.** All proteins identified from three independent mass-spec experiments.
DOI: https://doi.org/10.7554/eLife.37912.012
**Figure supplement 1.** Summary of mass-spec results.
DOI: https://doi.org/10.7554/eLife.37912.010
**Figure supplement 2.** Stage-specific expression of ERM-BP and confirmation by gel super-shift assay.
DOI: https://doi.org/10.7554/eLife.37912.011

*histolytica* clinical isolates that form cysts (*Ehrenkaufer et al., 2007*) as well as due to heat stress (*Hackney et al., 2007*).

In order to find out whether EIN_083100 is the ERM-BP, we overexpressed Myc-tagged EIN_083100 in *Entamoeba* trophozoites. The crude nuclear extract (NE) from Myc-tagged EIN_083100 overexpressing trophozoites showed a specific band in EMSA, while un-transfected trophozoite nuclear extract showed no band; these data confirm that the binding is due to overexpression of EIN_083100 (*Figure 2B*). Interestingly, EMSA with overexpressed Myc tagged-ERM-BP in trophozoites showed a faint second band, which may indicate that Myc tagged-ERM-BP binds to some factor present only in trophozoite nuclear extract or that NAD$^+$ level differences in trophozoites and cysts affect the binding.

We also performed gel super-shift assays using crude nuclear extract from Myc-tagged EIN_083100 overexpressing trophozoites, anti-Myc antibody and anti-actin antibody. EMSA with anti-Myc antibody resulted in a super-shift band (**) on gel shift assay, whereas EMSA with the

control actin antibody did not result in a super-shift *Figure 2—figure supplement 2B*. Taken together the data suggest that EIN_083100 is indeed an ERM-binding protein (ERM-BP).

## Myc-tagged ERM-BP is localized to nucleus and overexpression enhances encystation

In order to see the localization ERM-BP within the cell, we performed immunostaining on Myc-ERM-BP overexpressing cell lines. Immunofluorescence assay with anti-Myc-antibody in trophozoites revealed weak cytosolic staining and strong nuclear staining (*Figure 3A*). However, in 24 hr cysts Myc-ERM-BP localized exclusively to the nucleus (*Figure 3B*). Localization of ERM-BP to the nucleus supports the hypothesis that EIN_083100 is a transcription factor.

To further characterize ERM-BP roles in amebic biology, encystation efficiency was investigated in the ERM-BP overexpression cell line. Overexpression of ERM-BP resulted in a two-fold increase in

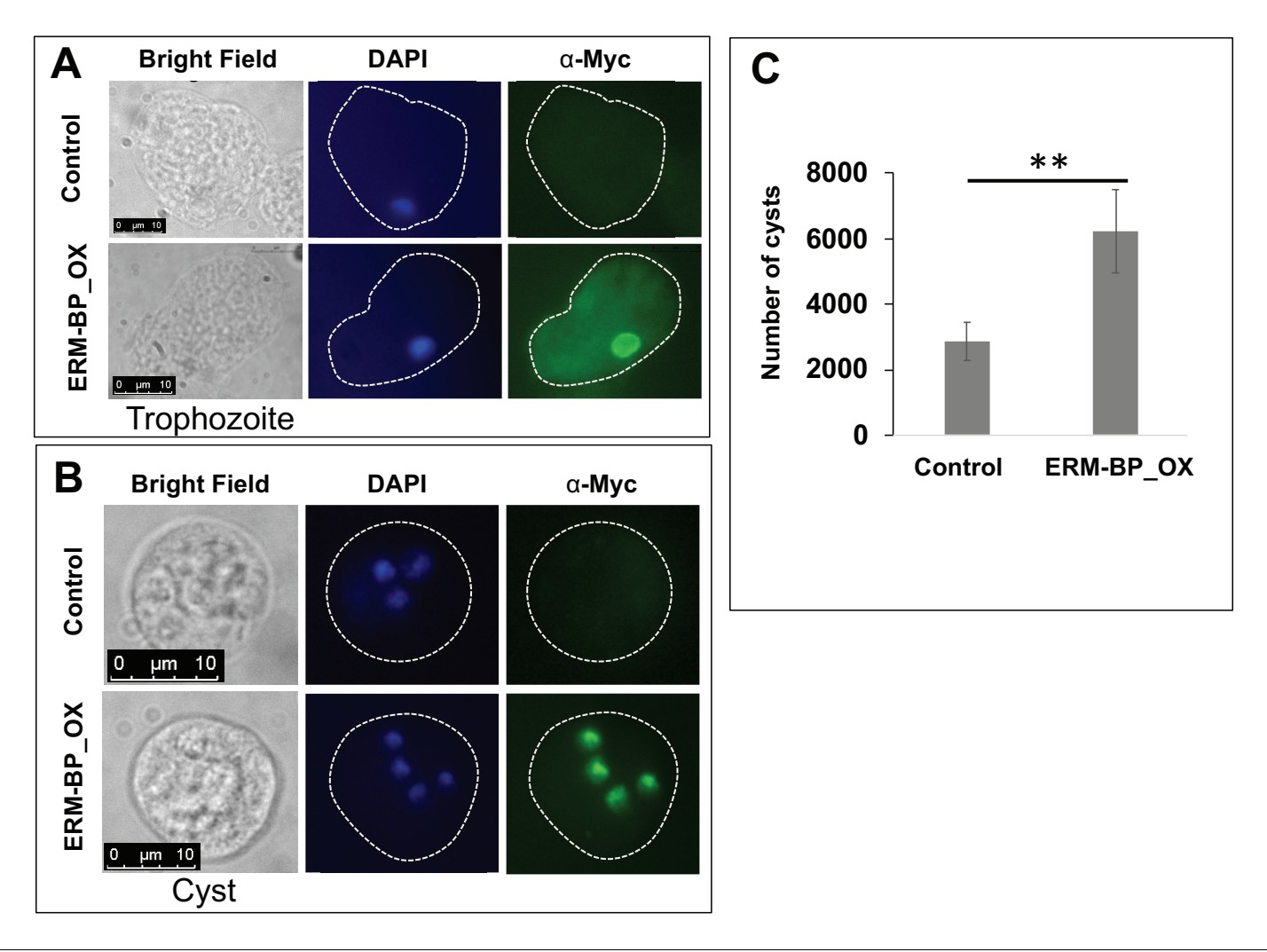

**Figure 3.** Myc-tagged ERM-BP is localized to nucleus and overexpression enhances encystation. Immunostaining with α-myc antibody in (A) trophozoites, and (B) 24 hr cyst was performed in ERM-BP_OX and control cells (Green). DNA was stained with DAPI (Blue). Scale bar for trophozoites and cysts are 10 μm. (C) Data represents the number of cysts in ERM-BP_OX and control cell lines after 72 hr of encystation. The number of cysts in control parasites was compared to that of ERM-BP_OX as determined by calcofluor staining and analyzed by ImageXpress (equipped with a laser and image-based acquisition) in a 96-well format. A minimum of eight wells per parasite line per experiment were analyzed, and biological replicate experiments were performed on three independent days. Data are mean ±s.e. (n = 3) Student's t-test; **p<0.001.
DOI: https://doi.org/10.7554/eLife.37912.013

cyst number compared to the control, suggesting that ERM-BP plays a role in regulation of encystation (*Figure 3C*).

## Silencing of ERM-BP expression levels reduces encystation efficiency and produces ghost like cysts

In order to better understand the role of ERM-BP, we used a trigger-mediated RNA-interference gene silencing approach to downregulate ERM-BP (*Suresh et al., 2016*). Transcript for EIN_083100 was undetectable in silenced ERM-BP cell lines at 24 hr of encystation, indicating successful silencing of ERM-BP (*Figure 4—figure supplement 1*). Loss of ERM-BP significantly decreased the cyst number in silenced ERM-BP cells (*Figure 4A*). Intriguingly, the cysts that were produced in the ERM-BP silenced cell line had altered morphology, including many 'ghost like' cysts with thin and irregular walls, suggesting a defect in cyst wall formation (*Figure 4B*). The synthesis of the *Entamoeba* cyst wall, as proposed by the wattle and daub model, occurs in three stages (*Chatterjee et al., 2009*). In the first stage or foundation stage, Jacob lectins bind to the plasma membrane Gal/GalNAc lectins; in the second or wattle stage, Jacob lectins cross-link the chitin fibrils that are deposited on the surface of encysting amoeba and finally, in the third or daub stage, the cyst wall is solidified by the addition of the Jessie-3 lectin resulting in an impermeable and rigid cell wall (*Chatterjee et al., 2009*). To further investigate whether there was any defect in cyst wall formation, we stained mature (72 hr) cysts with anti-Jacob antibodies. Disintegrated or uneven localization of Jacob was observed in cysts from ERM-BP silenced parasites, suggesting that there is indeed a defect in the cell wall formation when ERM-BP is silenced (*Figure 4C*). Quantitative analysis was done by staining the cyst wall using two markers: one with the calcofluor that stains the chitin in the cyst wall and a second with antibody against cyst wall protein Jacob. Using these two markers, we assessed the number of morphologically normal and defective cysts in the control and ERM-BP downregulated cell lines. About 65% of the silenced-ERM-BP cysts show defective cyst wall formation (*Figure 4D*) as compared to 16% of wild-type, control parasites. RT-PCR for Jacob (EIN_080730) showed no change in expression level in silenced ERM-BP compared to control during the encystation time points tested (*Figure 4—figure supplement 2A*) implying that the defect in Jacob-localization may be due to inadequacy in transport of cell wall components during encystation. Expression of chitinase (EIN_404540) and Jessie lectin (EIN_066080) is reduced in silenced ERM-BP cysts, suggesting that ERM-BP has an important role in the expression of a subset of cyst wall proteins (*Figure 4—figure supplement 2A*).

Additionally, we observed that silencing of ERM-BP decreased parasite viability in encystation media (*Figure 4E*). However, parasites with silenced ERM-BP had no obvious growth defects or altered viability under trophozoite conditions. This suggests that trophozoites which are starting to encyst but getting arrested in development have reduced viability. Next, we investigated the excystation of the ERM-BP silenced cells to see whether the cysts formed under ERM-BP silenced conditions can still excyst into functional trophozoites. We encysted ERM-BP silenced and control cells for 72 hr, treated overnight with water to lyse remaining trophozoites, inoculated equivalent numbers of cysts in excystation medium, and counted the number of trophozoites after 48 hr. Cysts from silenced ERM-BP parasites produced 3 – 4 fold fewer trophozoites compared to control cell lines, indicating that the defective cysts from ERM-BP silenced lines also fail to excyst normally (*Figure 4F*). Thus, overall, the downregulation of ERM-BP results in fewer cysts and even the cysts that are produced are largely defective in viability and ability to excyst.

## Gene expression changes that are dependent on ERM-BP

Our hypothesis is that ERM-BP is involved in the regulation of cyst-specific genes that have the ERM-motif in the promoter. To test this idea, we analyzed the expression of three putative target genes (EIN_095090, EIN_247000 and EIN_371110) all of which have ERM in their promoter, in ERM-BP overexpressed (ERM-BP_OX) trophozoites. RT-PCR detected expression of these target genes only in ERM-BP_OX trophozoites compared to control cells (*Figure 4—figure supplement 2B*). Further, we characterized the effect of ERM-BP silencing on expression of these target genes. RT-PCR was performed in cysts in which ERM-BP had been silenced for the same three target genes (EIN_095090, EIN_247000 and EIN_371110). RT-PCR showed reduced expression of all three target genes only in cysts in which ERM-BP had been silenced. However, in control cysts these target genes

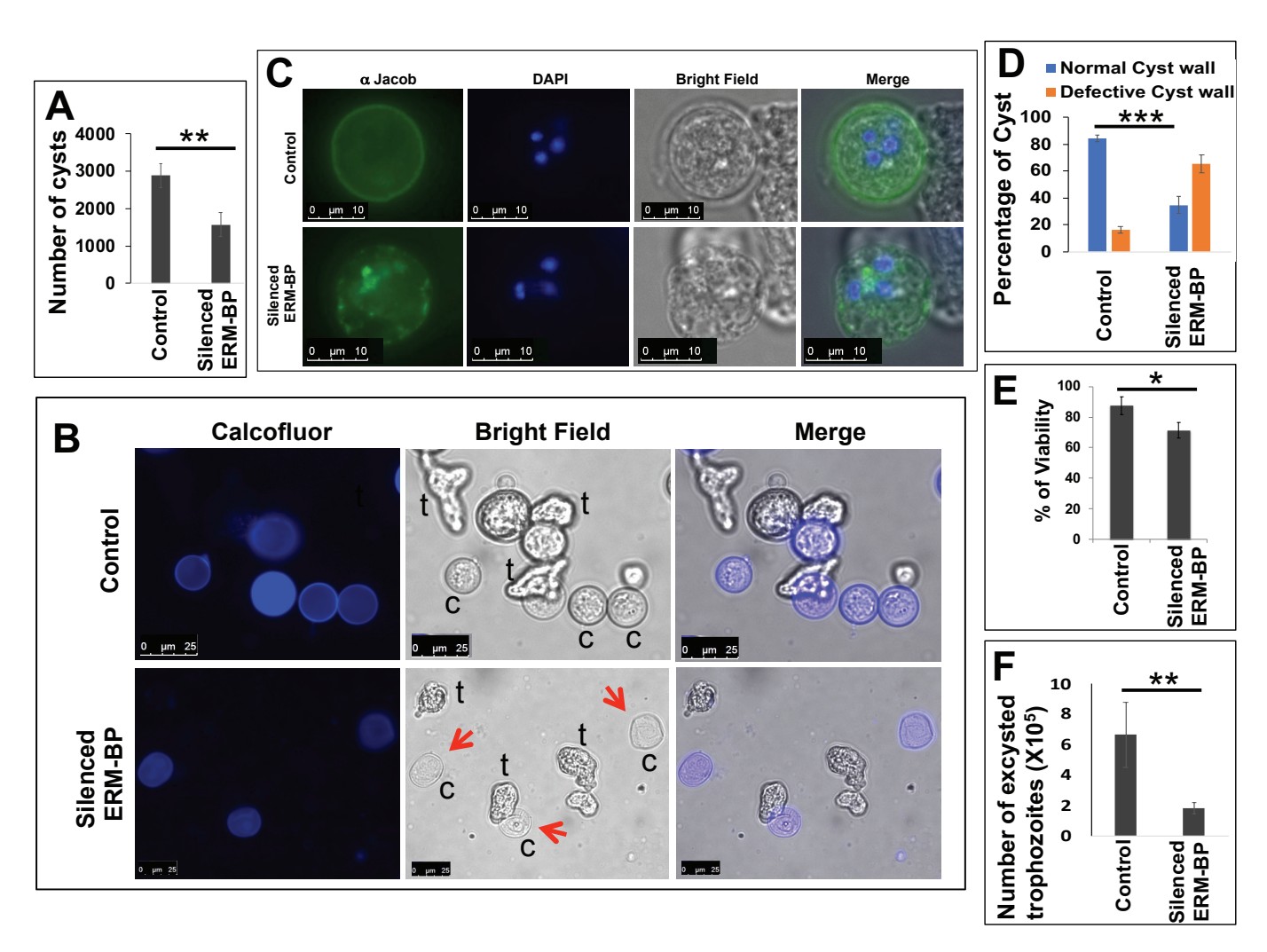

**Figure 4.** Silencing ERM-BP decreases encystation and leads to formation of ghost like cysts. (**A**) Data represents the number of cysts in control and silenced-ERM-BP cell lines after 72 hr of encystation. The number of cysts in control parasites was compared to silenced ERM-BP by calcofluor staining and analyzed by ImageXpress (equipped with a laser and image-based acquisition) in a 96-well format. A minimum of eight wells per parasite line per experiment were analyzed, and biological replicate experiments were performed on three independent days. Data are mean ± s.e. (n = 3) Student's t-test; **p<0.01. (**B**) Control and silenced-ERM-BP cells were encysted for 72 hr and cells were stained with calcofluor white (Blue) and imaged. Red arrows point to ghost like cysts; trophozoites are marked as 't' and cysts as 'c'. Scale bar, 25 μm. (**C**) Control and silenced-ERM-BP cysts (72 hr) were stained with anti-Jacob antibody followed by Alexa-488 conjugated secondary antibody (Green) and DNA was stained with DAPI (Blue). Scale bar 10 μm. (**D**) Percentage of cysts with defective cyst wall are determined by staining with anti-Jacob antibody in the control and ERM-BP silenced cell lines. Data are mean ±s.d. (n = 3) Student's t-test; ***p<0.002. (**E**) Percent viability of 72 hr encysted silenced-ERM-BP and control cells was determined by FDA fluorescence. Data are mean ±s.d. (n = 3) Student's t-test; *p<0.05. (**F**) The number of excysted trophozoites in silenced-ERM-BP and control cell lines. Data are mean ±s.d. (n = 3) Student's t-test; **p<0.01.

DOI: https://doi.org/10.7554/eLife.37912.014

The following figure supplements are available for figure 4:

**Figure supplement 1.** ERM-BP silencing using an RNAi-Trigger approach.
DOI: https://doi.org/10.7554/eLife.37912.015
**Figure supplement 2.** ERM-BP regulates the expression of target genes.
DOI: https://doi.org/10.7554/eLife.37912.016

showed differential but comparatively higher expression compared to the silenced ERM-BP cell line. Taken together, these data suggest that the expression of these genes is regulated by ERM-BP (*Figure 4—figure supplement 2C*).

When we analyze the 131 genes having ERM in the promoter, we identified that 65 (~50%) are annotated as hypothetical. Of the remainder, some genes are annotated as cyst wall proteins (EIN_404540: Chitinase; EIN_066080: Jessie lectin; EIN_284810: Chitin synthase) (*Figure 1—source data 3*). There are also few genes having ERM in the promoter that are associated with metabolism and stress response. However, considering 50% are hypothetical genes, it is difficult to speculate whether any specific pathways are regulated by ERM-BP (*Figure 1—source data 3*). Further analysis and global understanding of the regulatory role of ERM-BP in *Entamoeba* gene expression can be performed in the future by the analysis of RNA-Seq data from ERM-BP_OX and ERM-BP silenced cell lines.

## Intracellular NAD$^+$/NADH is elevated during encystation and NAD$^+$ facilitates encystation

The ERM-BP does not possess any canonical DNA-binding domain but has a cysteine hydrolase superfamily domain at the C-terminus, which resembles bacterial nicotinamidase and may bind to NAD$^+$. We observed that NAD$^+$/NADH ratio is significantly elevated during encystation (*Figure 5A*). Our findings showed that during encystation the NAD$^+$ level gradually increased and the NADH level decreased, which was consistent in all the experiments using two different assay methods (NAD$^+$/NADH assay kit (Abcam) and NAD$^+$/NADH-Glo assay kit (Promega). The actual concentrations of NAD$^+$ and NADH measured in trophozoites were 1.5 ± 0.34 and 2.3 ± 0.75 nmoles respectively per $2 \times 10^6$ cells; in 24 hr encysted cells the NAD$^+$ and NADH concentrations were 4.1 ± 1.40 and 1.2 ± 0.40 nmoles per $2 \times 10^6$ cells, respectively. Additionally, we measured the NAD$^+$ and NADH concentration in sarkosyl-resistant cysts and saw a similar elevated amount of NAD$^+$ in the mature, sarkosyl-resistant cysts (*Figure 5—figure supplement 1*).

In earlier studies Jeelani et al., reported that NAD$^+$ levels slightly increased in the earlier time-point (i.e. 2 hr) of encystation, but that in later time points of encystation the NAD$^+$ level gradually decreased (*Jeelani et al., 2012*). However, the change in NADH level was not measured. Our current observations do not match the previous report and for this there are multiple possible explanations. First, Jeelani et al., used 75% chilled methanol to quench the metabolic activity. This also fixes the cells and NAD$^+$ may leach out during this fixation. Thus, in the later steps when the metabolites are extracted with chloroform and water by sonication to lyse cysts, the results may not accurately measure the NAD$^+$ levels. Second, NAD$^+$ is very unstable and can be easily reduced to NADH and this may lead to an apparent artificial reduction of NAD$^+$. Despite the differences between our results and Jeelani et al., our results are highly internally consistent, using two assay methods, measuring both NAD$^+$ and NADH and measuring NAD$^+$ in mature (sarkosyl resistant) cysts.

Intracellular NAD$^+$/NADH levels control the activity of transcription factors in other systems (*Rutter et al., 2001*; *Ravcheev et al., 2012*) and we hypothesized that the elevated NAD$^+$ level may have a role in regulating encystation. To explore whether extracellular NAD$^+$ has any role in encystation, we added 1 mM extracellular NAD$^+$ to the encysting parasites and found a significant enhancement of encystation. Addition of extracellular NAD$^+$ increased the cyst number more than two-fold compared to untreated parasites (*Figure 5C*). We demonstrated that exogenously applied NAD$^+$ can be taken into *Entamoeba* cells by measuring the intracellular NAD$^+$ concentration in parasites after exposure to exogenous NAD (*Figure 5B*). Our results show a gradual increase of intracellular NAD$^+$ in *Entamoeba* cells treated with 1 mM of NAD$^+$ up to 8 hr; however, despite the large amount of extracellular NAD$^+$ concentration the intracellular NAD$^+$ remained in nmole range (*Figure 5B*). Of note, NAD$^+$ in millimolar concentration is often used in experiments to determine the effect of exogenous NAD$^+$ (*Billington et al., 2008*) and in some systems endogenous intracellular NAD$^+$ can be in millimolar range (*Ying et al., 2003*). Furthermore, we observed an additive role of NAD$^+$ and overexpression of ERM-BP in encystation. The addition of NAD$^+$ to ERM-BP overexpressed cells further enhanced encystation efficiency. However, silenced ERM-BP parasites did not show any enhancement in encystation by the addition of NAD$^+$, suggesting that the effect of NAD$^+$ in influencing encystation efficiency is via ERM-BP (*Figure 5C*).

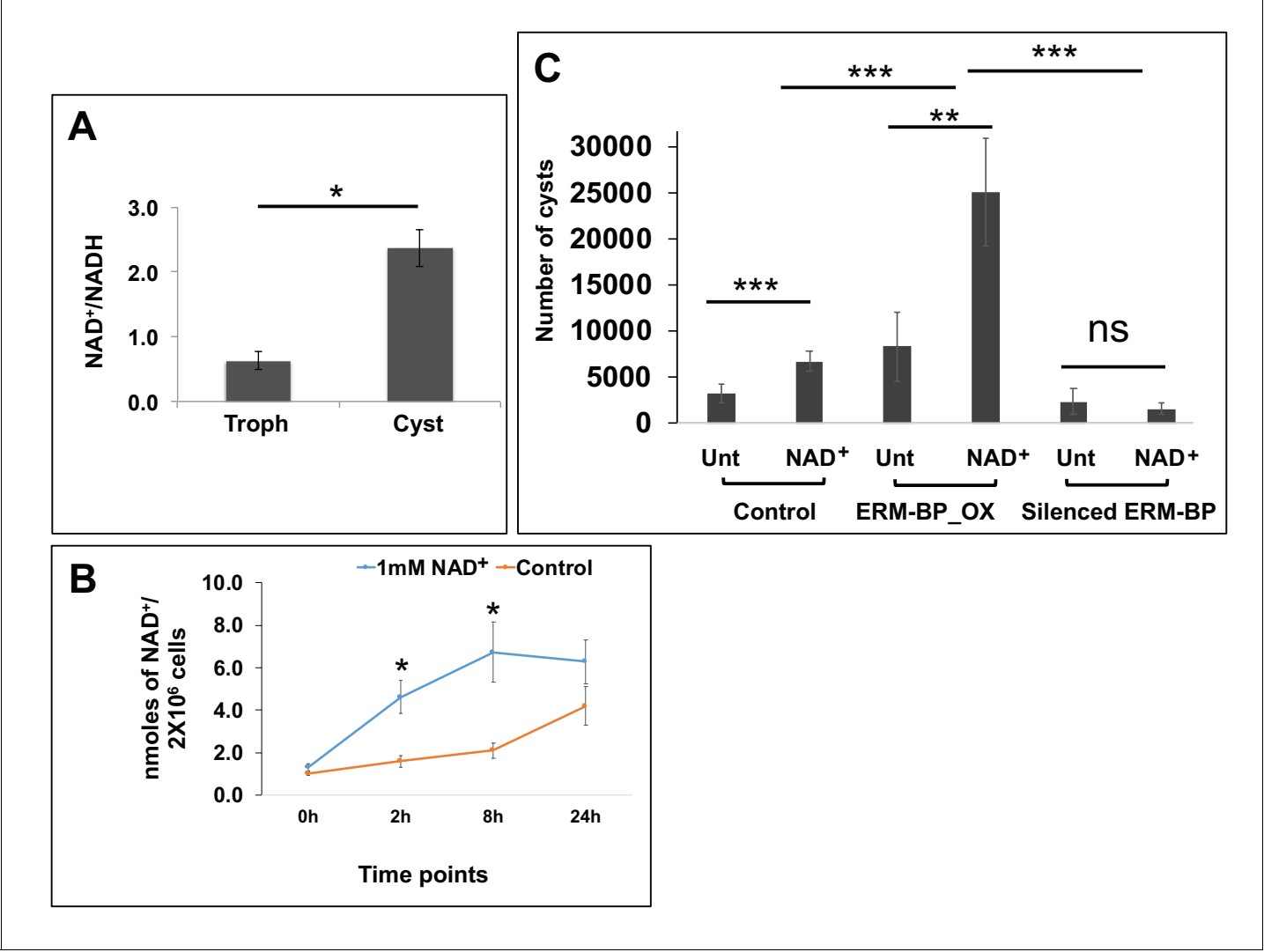

**Figure 5.** Intracellular NAD$^+$/NADH is elevated during encystation and NAD$^+$ facilitates encystation. (**A**) Measurement of intracellular NAD$^+$/NADH in trophozoites (Troph) and 72 hr of cysts. Data are mean ±s.d. (n = 3) Student's t-test; *p<0.05. (**B**) Intracellular NAD$^+$ was detected from *Entamoeba* cells treated with 1 mM of NAD$^+$ or untreated control at different time points. Data represents mean ±s.d. (n = 3) Student's t-test; *p<0.05. (**C**) The number of cysts in 1 mM NAD$^+$ treated (NAD$^+$) or untreated (Unt) Control, ERM-BP_OX and Silenced ERM-BP cells are shown. Data are mean ±s.d. (n = 3) Student's t-test; **p<0.01, ***p<0.001.

DOI: https://doi.org/10.7554/eLife.37912.017

The following figure supplements are available for figure 5:

**Figure supplement 1.** Intracellular NAD$^+$ and NADH in trophozoites and sarkosyl resistant cysts.

DOI: https://doi.org/10.7554/eLife.37912.018

**Figure supplement 2.** NAD$^+$ but not NADH facilitates DNA binding.

DOI: https://doi.org/10.7554/eLife.37912.019

## Domain characterization, NAD$^+$/DNA binding and catalytic activity of ERM-BP

We performed gel shift assays using recombinant ERM-BP in presence of variable amounts of NAD$^+$ to investigate its role in DNA binding. Our EMSA results show that addition of NAD$^+$ facilitates DNA-binding activity of bacterially expressed recombinant ERM-BP (*Figure 5—figure supplement 2*). We also observed that the increase in DNA binding activity is very specific to addition of NAD$^+$ as addition of NADH did not increase DNA-binding activity; rather higher concentrations of NADH decreased DNA binding (*Figure 5—figure supplement 2*). Given that this observation was with

bacterially expressed recombinant protein, it implies that other amebic-cellular factors are not necessary for this interaction. It is plausible that $NAD^+$ directly binds to ERM-BP inducing conformational changes to facilitate DNA-binding activity. However, we did not identify any typical $NAD^+$-binding domain (Gly-X-Gly-XX-Gly) (*Sganga and Bauer, 1992*) in ERM-BP raising the possibility that ERM-BP binds $NAD^+$ through the nicotinamidase domain.

Alignment of ERM-BP protein with two bacterial nicotinamidase PncA (*Mycobacterium tuberculosis* and *Acinetobacter baumanii*) identified specific conserved residues involved in nicotinamide binding (C198) and nicotinamide catalysis (C198, D12 and K150) (*Zhang et al., 2008*). These residues are conserved in ERM-BP from multiple *Entamoeba* species (*Figure 6*).

In order to investigate the $NAD^+$-binding sites in ERM-BP, we generated a number of mutants: C198A (which lost both nicotinamide binding and catalysis activity in bacterial PncA), and two mutants D12A and K150A (which lost catalysis activity in bacterial PncA) (*Zhang et al., 2008*). Additionally, although our in vitro results confirmed that purified ERM-BP specifically binds to the 'CAA-CAAA' DNA motif, we noted that ERM-BP does not possess any canonical DNA-binding domain. Bioinformatics analysis by using BindN (*Wang and Brown, 2006*), a web-based efficient prediction of DNA-binding sites in amino acid sequence predicted clusters of amino acids ($S_AR_LTKR$) in ERM-BP as shown by arrowheads in *Figure 6*, which may be involved in DNA binding. To test this prediction, we generated an ERM-BP-DNA Binding Mutant (ERM-BP-DBM) by changing all five predicted amino acids to alanine (marked as black arrowhead in a red box in *Figure 6*). We analyzed WT and mutant ERM-BP by using three approaches: (i) DNA binding as determined by EMSA analysis in the presence and absence of $NAD^+$, (ii) $NAD^+$-binding efficiency as assayed by protein thermal shift assay, and (iii) catalytic activity by measuring conversion of nicotinamide to nicotinic acid by HPLC. The results are summarized in *Table 2*.

EMSA was performed using ERM-BP-WT and recombinant proteins in the presence of different concentrations of $NAD^+$. Mutants D12A and K150A bound DNA as measured by EMSA band shift, and the DNA-binding increased with increasing $NAD^+$ concentration, though to a lesser extent compared to ERM-BP-WT; however, C198A did not bind DNA (*Table 2*). The ERM-BP-DBM shows a significant reduction in DNA-binding activity even at higher concentrations of $NAD^+$ (*Figure 6—figure supplement 1*).

In order to more directly assess the interaction between ERM-BP and $NAD^+$, we performed protein thermal shift assay, which measures the change in melting temperature (Tm) when a ligand binds to and stabilizes a protein (*Soon et al., 2012*; *Miyazaki et al., 2017*). The ERM-BP-WT showed a significant thermal shift with increasing $NAD^+$ concentrations indicating protein-ligand binding; the three mutants D12A, K150A and ERM-BP-DBM also had similar Tm values indicating protein-ligand binding (*Table 2*). However, the C198A mutant did not have a thermal shift with increasing concentrations of $NAD^+$, suggesting that C198 is necessary for $NAD^+$ binding. We also observed that the $NAD^+$ and ERM-BP binding is very specific as NADP did not result in a protein thermal shift in the Tm (*Figure 6—figure supplement 2*). All Tm values are shown in *Table 2* and the melting curves are shown in *Figure 6—figure supplement 2*..

In order to characterize the catalytic activities of ERM-BP, HPLC analysis was performed. ERM-BP-WT and ERM-BP-DBM showed catalytic activity and conversion of nicotinamide to nicotinic acid (40–50% conversion observed). In contrast, all three ERM-BP mutants that had mutations in the nicotinamidase domain (C198A) or in the other two conserved residues (D12A and K150A) were unable to process nicotinamide showing that all three conserved residues are important for catalysis (*Table 2*) (*Figure 6—figure supplement 3*).

The results of the DNA binding, $NAD^+$ binding, and nicotinamide catalysis are summarized in *Table 3*. The data demonstrate that binding to DNA and $NAD^+$ are linked; mutations that decrease $NAD^+$ binding also decrease DNA binding (C198A). However, DNA binding and enzymatic conversion of nicotinamide can be separated, as shown by the activities of the D12A and K150A mutants. Mutants that abolish DNA binding have no effect on the catalytic activity (ERM-BP-DBM), whereas catalytically deficient mutants retain DNA-binding properties (D12A and K150A). The C198A mutation has the greatest effect on ERM-BP function and abolishes both $NAD^+$ and DNA binding and nicotinamide turnover. Taken together the data suggest a model for the role of ERM-BP in encystation (*Figure 7*). $NAD^+$-binding changes the conformation of ERM-BP and facilitates ERM-BP binding to DNA. This binding of ERM-BP to the promoters of cyst-specific genes directly increases expression of genes in the developmental cascade. The residues most important for this activity are C198 and

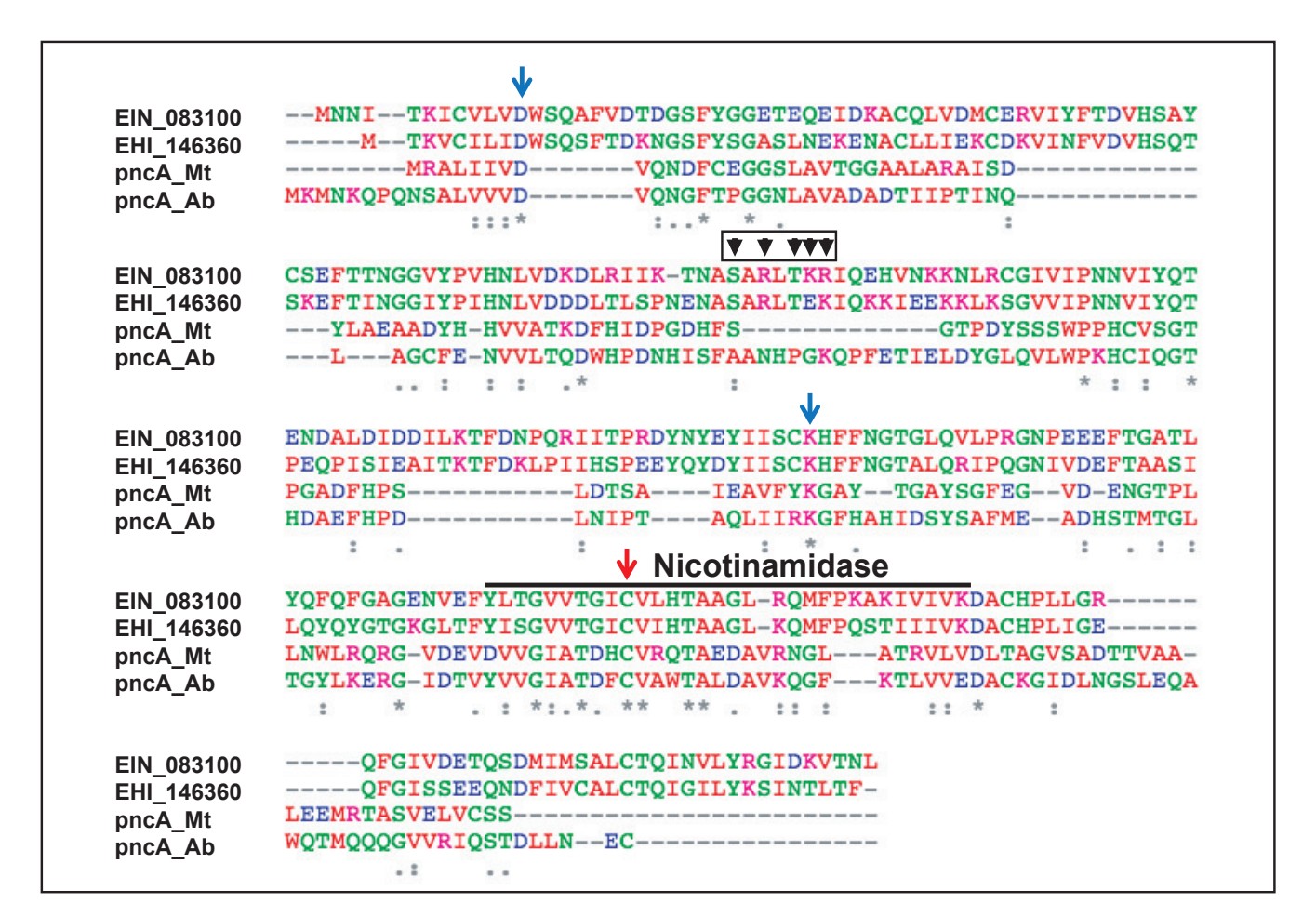

**Figure 6.** Protein sequence alignment and residues in ERM-BP important for NAM binding and catalytic activity. Protein sequence alignment of ERM-BP from two *Entamoeba* species (*E. invadens*- EIN_083100 and *E. histolytica*- EHI_146360) and bacterial nicotinamidase (pncA) from *Mycobacterium tuberculosis* (Mt) and *Acinetobacter baumanii* (Ab) are shown. The nicotinamidase domain is underlined and the key residues involved in nicotinamide (NAM) binding and catalysis are shown by arrows. Mutants are generated by changing each relevant residue into an Alanine. The mutants are D12A, K150A, and C198A. The predicted DNA-binding region and residues are shown by arrowheads with a red box as five amino acids cluster ($S_A R_L TKR$) except subscripts-A and L. An ERM-BP-DBM was generated by changing all five predicted amino acids into Alanine using the Gibson assembly method.

DOI: https://doi.org/10.7554/eLife.37912.020

The following figure supplements are available for figure 6:

**Figure supplement 1.** EMSA with recombinant proteins in the presence of varying concentrations of NAD[+].
DOI: https://doi.org/10.7554/eLife.37912.021
**Figure supplement 2.** Thermal stability of ERM-BP-WT and mutants in presence of different concentrations of NAD[+].
DOI: https://doi.org/10.7554/eLife.37912.022
**Figure supplement 3.** Enzymatic activity of ERM-BP-WT and mutants.
DOI: https://doi.org/10.7554/eLife.37912.023

the cluster of five amino acids (SRTKR) in the N-terminal of ERM-BP (*Table 3*). Additionally, ERM-BP mediates catalysis of nicotinamide to nicotinic acid. Nicotinic acid may function as a second messenger or be converted into NAD[+] by a salvage pathway to mediate stage conversion. Our results find that the residues most important for this enzymatic activity are C198, D12 and K150 (*Table 3*).

**Table 2.** DNA binding, NAD$^+$ binding and catalytic activity of ERM-BP-WT and ERM-BP mutants.
Wild type (WT) and mutant ERM-BP proteins were analyzed by three approaches. DNA binding was determined by EMSA analysis with increasing concentration of NAD$^+$. Protein thermal shift assay was performed to determine NAD$^+$ binding and melting temperature (Tm) in the presence of varying concentration of NAD$^+$ is shown. Nicotinamidase activity was determined by measuring conversion of nicotinamide to nicotinic acid by HPLC.

| Proteins | DNA Binding (EMSA) NAD$^+$ (mM)# | | | NAD+ Binding (Tm) NAD$^+$ (mM)§ | | | | Nicotinamidase activity (% conversion)¶ |
|---|---|---|---|---|---|---|---|---|
| | 0 | 1 | 4 | 0 | 1 | 4 | P-values (0-4) | |
| 1. GST† | - | - | - | 52 ± 7.0 | 53 ± 7.0 | 51 ± 7.0 | 0.6549 | no conversion |
| 2. ERM-BP-WT‡ | ++ | +++ | ++++ | 57 ± 0.3 | 60 ± 0.7 | 62 ± 1.3 | 1.1E-08 | 52 ± 0.6 |
| 3. D12A‡ | +/- | ++ | +++ | 57 ± 1.0 | 59 ± 1.0 | 62 ± 1.0 | 1.5E-08 | no conversion |
| 4. ERM-BP-DBM‡ | - | - | ++ | 57 ± 0.3 | 58 ± 0.8 | 61 ± 1.0 | 9.9E-09 | 50 ± 2.1 |
| 5. K150A‡ | +/- | ++ | +++ | 58 ± 1.7 | 59 ± 1.0 | 62 ± 11 | 3.7E-05 | no conversion |
| 6. C198A† | - | - | -/+ | 57 ± 0.3 | 57 ± 0.4 | 57 ± 0.5 | 0.3796 | no conversion |

# EMSA (Electrophoretic mobility shift assay) with varying amounts of NAD$^+$ (0, 1, 4 mM): '++++' indicates strong binding, '+++' and '++' indicate moderate, '+/-' indicates weak binding and '-'"indicates no binding. § NAD$^+$ binding was monitored by protein thermal shift assay with varying amounts of NAD$^+$ (0, 1, 4 mM). The Tm (melting temperature) is shown as mean ±s.d. (n = 3) Student's t-test; p-values are shown between NAD$^+$ 0 mM and 4 mM concentrations. ¶ The turnover of nicotinamide to nicotinic acid by ERM-BP-WT and mutant recombinant proteins are shown as percentage conversion mean ±s.d. (n = 3). ‡ Indicates having significant effects in DNA /NAD$^+$ binding and catalytic activity, † Indicates not having significant effects in DNA/ NAD$^+$ binding and catalytic activity.

DOI: https://doi.org/10.7554/eLife.37912.024

## Discussion

In this study, we identified a novel transcription factor ERM-BP as a key regulator of stage conversion in the protozoan parasite *Entamoeba*. The ERM-BP transcription factor regulates the formation of mature cysts by directly controlling promoter occupancy. The catalytic role of ERM-BP on nicotinamide to produce nicotinic acid may also have important roles in encystation. The function of ERM-BP is modulated by changes in intracellular NAD$^+$ levels, thus linking cellular energetics to a molecular mechanism controlling parasite development. Based on our data, we have developed a model for the role of ERM-BP and its effect on development and regulation by NAD$^+$ interaction (*Figure 7*).

The activity of ERM-BP is regulated by intracellular levels of NAD$^+$. In multicellular eukaryotes, the redox state of NAD$^+$ modulates the activities of NPAS2: BMAL1 transcription factors and also controls the circadian clock-associated gene expression in mammals (*Rutter et al., 2001*). Transcriptional regulation of central carbon and energy metabolism by redox-responsive repressor Rex in bacteria is regulated by intracellular level of NAD$^+$ (*Ravcheev et al., 2012*). Calorie restriction extends

**Table 3.** Summary of DNA binding, NAD binding and enzymatic activity of wild type and mutant versions of ERM-BP are shown.
DNA binding, NAD$^+$ binding and enzymatic activity of ERM-BP-WT and ERM-BP mutants are summarized. '+' indicates binding or having enzymatic activity and '-' indicates no binding or no activity.

| Proteins | DNA Binding | NAD$^+$ binding | Enzymatic activity |
|---|---|---|---|
| ERM-BP-WT | + | + | + |
| D12A | + | + | - |
| ERM-BP-DBM | - | + | + |
| K150A | + | + | - |
| C198A | - | - | - |

DOI: https://doi.org/10.7554/eLife.37912.025

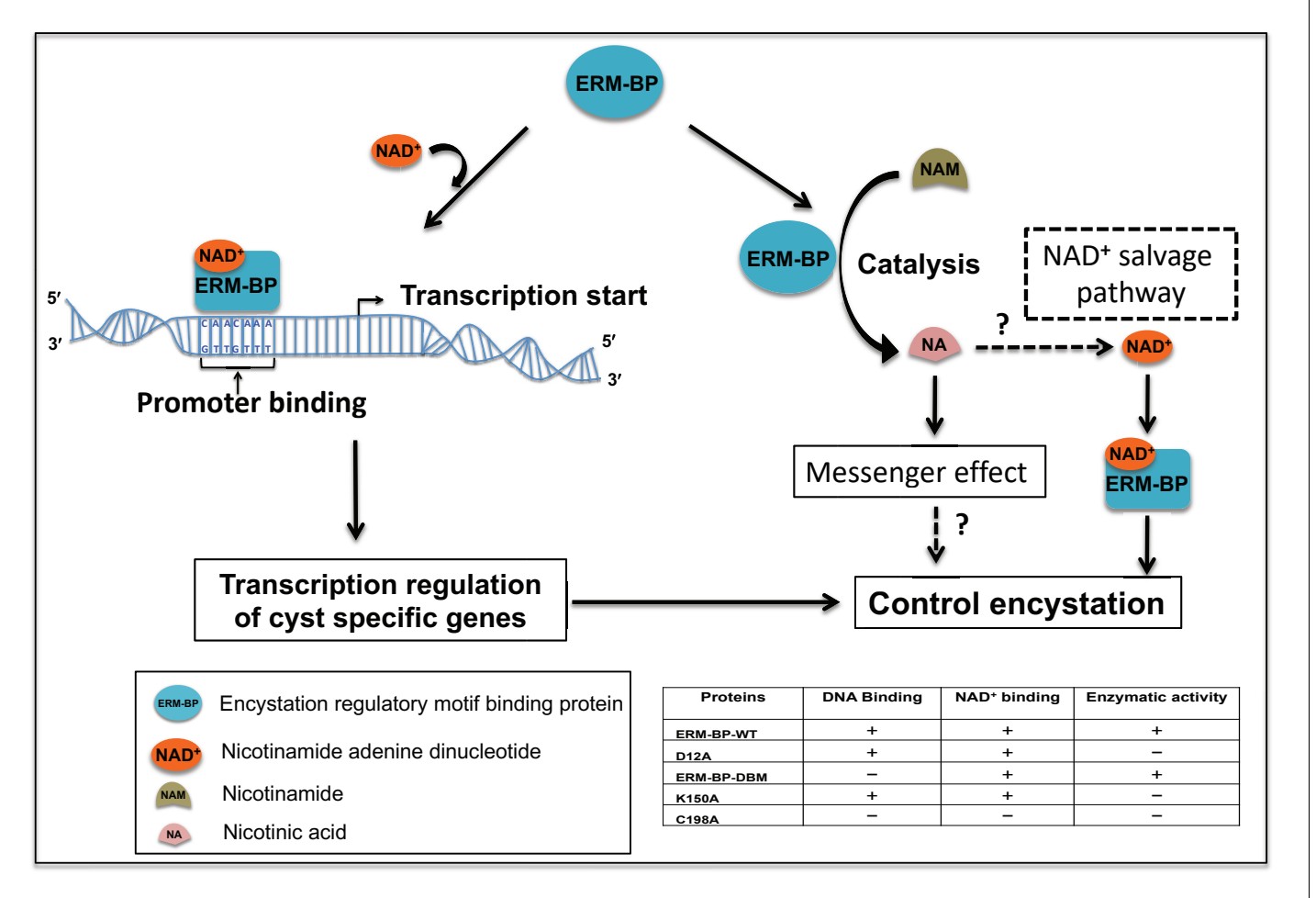

**Figure 7.** Proposed model for the role of ERM-BP in encystation. The transcription factor ERM-BP binds to NAD$^+$ and undergoes a conformational change, which facilitates the binding to the CAACAAA motif in the promoter of cyst-specific genes. ERM-BP also catalyzes nicotinamide (NAM) into nicotinic acid (NA), which may act as a second messenger and might have role in encystation. Another possibility is that nicotinic acid may be converted back to NAD$^+$ via a salvage pathway and can then bind to ERM-BP and regulate encystation. Together these functions of ERM-BP control the encystation process in *Entamoeba*.

DOI: https://doi.org/10.7554/eLife.37912.026

life span in a wide variety of species and increases the replicative life span in yeast by activating Sir2, a highly conserved NAD$^+$-dependent deacetylase (*Lin et al., 2004*). NAD$^+$ is also a putative metabolic regulator of transcription, longevity and several age-associated diseases, including diabetes, cancer and neurodegenerative diseases (*Garcia-Soriano et al., 2001*; *Greenamyre et al., 2001*; *Lin and Guarente, 2003*; *Zhang et al., 2008*). In the soil amoeba *Dictyostelium discoidium*, the NAD$^+$/NADH levels change significantly during development (*Bredehorst et al., 1980*). It is also reported that in the malaria parasite *Plasmodium falciparum* elevated NAD$^+$ levels play a crucial role in blood stage development (*O'Hara et al., 2014*).

In *Entamoeba* nutrient restriction, which occurs by glucose depletion and osmotic stress, acts as a catalyst that induces the encystation process. During encystation, the levels of a majority of metabolites involved in glycolysis markedly decrease (*Jeelani and Nozaki, 2014*). Pyruvate is converted to acetyl-CoA by pyruvate:ferredoxin oxidoreductase, and acetyl-CoA is either converted to acetate with a concomitant ATP generation or reduced to ethanol with regeneration of NAD$^+$ (*Jeelani and Nozaki, 2014*). Our results show not only the augmentation of NAD$^+$ levels, but also that NADH levels decrease during encystation. Thus, there is an overall enrichment of the intracellular NAD$^+$/NADH ratio during *Entamoeba* encystation. Given these links between the changing energy state of

the parasite and stage conversion, it is reasonable to hypothesize that changing levels of metabolic cofactors could be important in triggering encystation.

The enrichment of intracellular $NAD^+$/NADH during *Entamoeba* encystation and enhancement of ERM-BP binding to the CAACAAA motif by $NAD^+$ in vitro support the notion that $NAD^+$ regulates ERM-BP function. In addition, our data demonstrate that $NAD^+$ plays a crucial role by binding to ERM-BP, changing its confirmation and facilitating ERM-BP binding to the promoter motif (*Figure 7*). Mutational analysis of the ERM-BP reveals that $NAD^+$ binding and DNA binding are linked (*Table 3*). Taken with the finding that $NAD^+$ binding changes protein thermal stability, it implies that $NAD^+$ binding induces a conformational change in ERM-BP, which then allows ERM-BP to bind to DNA.

A second, apparently distinct, function of ERM-BP is the enzymatic catalysis and nicotinamidase activity that generates nicotinic acid from nicotinamide, which may be acting as a second messenger. Nicotinic acid has been reported to control the expression of several genes including virulence factors in bacterial system, and serine threonine kinase in insulin-sensitive animal tissues and plays an important role in gene expression (*Choi et al., 2011*; *Edwards et al., 2013*). Our results suggest this catalytic activity is independent of DNA binding as demonstrated by the D12A, K120A, and ERM-BP-DBM mutants. The C198A mutation in the nicotinamidase domain is the only mutation that abolishes both DNA binding and nicotinamide catalysis. Which of these two functions of ERM-BP is more important for regulating encystation-specific gene expression is not clear at present, but future analysis with the specific mutant proteins will allow further dissection of this observation.

ERM-BP specifically binds to a consensus promoter motif 'CAACAAA' and regulates the expression of a set of cyst-specific genes. ERM-BP is highly conserved in all *Entamoeba* species that can form cysts but appears to be unique to *Entamoeba* as it lacks homologs in other parasites or in other eukaryotes. We have previously identified a Myb transcription factor that binds to a CCCCCC promoter motif and controls stage conversion in *Entamoeba* (*Ehrenkaufer et al., 2009*). ERM-BP represents the second transcription factor known to control the developmental pathway in *Entamoeba*. Future work with ERM-BP will involve identifying the transcriptional network it regulates as well as key interacting protein partners.

Stage conversion is a basic biological process, central to disease transmission and pathogenesis for *Entamoeba*. Identification of key regulators of this pathway provides molecular insights into the developmental cascade, but also identifies potential measures to target to block stage conversion. Future efforts can focus on ERM-BP and other regulators of development as targets for drug development to prevent disease transmission.

## Materials and methods

### Parasite culture, transfection and induction of stage conversion

*E. invadens* (strain IP-1) was axenically maintained (*Clark and Diamond, 2002*). To make stable transgenic cell lines, parasites were transfected with plasmid DNA by electroporation (*Ehrenkaufer and Singh, 2012*). The stable cell lines were maintained at G418 concentration of 80 µg/mL unless otherwise stated. To induce encystation, *E. invadens* trophozoites were incubated in 47% LYI-LG (supplemented with 7% adult bovine serum) in a 96-well plate (*Sanchez et al., 1994*). The cyst number was determined by automated quantitative imaging as described in earlier studies (*Suresh et al., 2016*; *Ehrenkaufer et al., 2018*). Briefly, calcofluor white, which specifically stains chitin in the cyst wall, was added to wells after 48 hr or 72 hr of encystation. The calcofluor stained cysts were imaged at 10X magnification using ImageXpress Micro (Molecular Devices) and quantified by using MetaXpress analysis software (Molecular Devices). The experiment was repeated at least three times with eight replicates for each sample. Data are represented as mean with standard errors and the t-test was performed from well distributed data set (24 replicates) of each cell line. For excystation experiments, trophozoites were encysted for 72 hr and any remaining trophozoites lysed by incubating overnight (~16 hr) in distilled water at 4°C. The remaining cysts were induced to excyst by incubating $8 \times 10^5$ cysts in LYI-LG media supplemented with 1 mg/mL bile, 40 mM sodium bicarbonate, 1% glucose and 10% serum for 48 hr (*Mitra et al., 2010*). Excystation was done in duplicate tubes for each sample and the efficiency was determined by counting trophozoites in a haemocytometer.

## Viability assay

Control and silenced ERM-BP cells were encysted in eight replicates in a 96-well plate. The viability was determined after 72 hr of encystation by Fluorescein Diacetate (FDA). The cells were treated with FDA in encystation media at 20 µg/mL final concentration for 30 min. Media was removed, washed once with 1X PBS and 100 µl 1X PBS was added. Fluorescence was measured by using a Tecan plate reader with an excitation and emission at 490 nm and 525 nm, respectively. The experiment was repeated on 3 different days; a Student's t-test was used to determine the p values.

## Bioinformatics analysis to identify consensus DNA promoter motifs

500 nt of the upstream promoter regions of 616 cyst specific genes were analyzed to identify DNA motifs as described earlier (*Hackney et al., 2007*; *Pearson et al., 2013*). In brief, MEME was performed with the command line: -dna -mod zoops -minw 6 -maxw 15 -minsites 5 -nmotfs 30. The MAST program was utilized to determine the total number of occurrences of each motif in the promoter sequence databases. The hypergeometric distribution was used to determine the significance of enrichment for each motif identified. Motifs with p-value of less than 0.001 were determined to be enriched significantly within the promoters of the cyst-specific genes. Sequence logos were generated using the R package SeqLogo.

## Electrophoretic mobility shift assays (EMSA)

EMSA was performed as previously described (*Pearson et al., 2013*). The oligonucleotides used in EMSA are listed in *Figure 1—source data 2*. Each motif had additional 12-nt at 5' and 8-nt at 3', which creates a 5'-overhang after annealing and utilized for radiolabeling using Klenow. In brief, complementary overlapping probes were annealed and labeled using [$^{32}$P] α-ATP and Klenow fragment (Invitrogen)(*Hackney et al., 2007*). Binding reaction was set in a total volume of 20 µl, which included 2 µl 10X EMSA-binding buffer (10 mM Tris-HCl, pH 7.9, 50 mM NaCl, 1 mM EDTA, 3% glycerol, 0.05% milk powder, and 0.05 mg of bromophenol blue), 5 µg of nuclear extract form trophozoites or 24 hr cysts, 2 µg of poly (dI-dC), and 50 fmol of labeled probe. The binding reaction mix was incubated for 30 min at room temperature, and samples were loaded onto a 9% non-denaturing polyacrylamide gel and run for 3 hr. The gel was fixed, dried, and exposed to a phosphor screen. Gels were imaged using Personal Molecular Imager (PMI) System with Quantity One software, Bio-Rad.

## Mutational analysis to identify critical residues of ERM

ERM is comprised of seven nucleotides with the consensus sequence CAACAAA. Three sets of mutations generated by changing the conserved nucleotides as shown *Figure 1C*. All the oligonucleotides used in EMSA are listed in *Figure 1—source data 2*. The cyst-specific promoter construct, C4, has a WT-ERM, a luciferase reporter gene and 3' regulatory region (*Manna et al., 2014*). ERM was mutated by changing CA to TG at fourth and fifth position by site-directed mutagenesis kit (Agilent-QuikChange Lightning). Mutations at the intended ERM nucleotides were confirmed by sequencing.

## Luciferase assays

Luciferase assays were performed a minimum of three times as described earlier (*Morf et al., 2013*). Briefly, trophozoites were chilled, harvested and washed once with 1X PBS, re-suspended in 1X lysis buffer (luciferase assay system, Promega E1500) complemented with protease inhibitors (1X HALT protein inhibitor cocktail, 1X E64), incubated on ice for 30 min to lyse trophozoites and spun down at 14,000 rpm for 15 min. Cysts were lysed by sonication (five pulses at 15 amp for 15 s). Protein concentration was measured by Bradford assay. A total of 30 µg of protein was added to luciferase reagent (Promega) and luciferase activity was measured by a luminometer as described earlier (*Morf et al., 2013*).

## Mass spectrometry identification of ERM-BP

To identify the ERM-binding protein, we performed LC-MS excising the shifted band from EMSA gel. EMSA was performed by using ERM-WT and ERM-core as control with cyst nuclear proteins. Gel slices were excised from the unlabeled lanes of ERM-WT and ERM-core from three biological

experiments. In one experiment, EMSA was performed by using ERM-WT and trophozoite nuclear extract, which served as a separate control. Gel slices were reduced with 5 mM DTT and alkylated with acrylamide followed by digestion by using Promega MS grade trypsin overnight as reported previously, with the addition of the acid-labile surfactant ProteaseMAX (Promega) (*Shevchenko et al., 2006*; *Pearson et al., 2013*). Peptides were extracted and dried prior to reconstitution and analysis. The mass spectrometry was performed by using an LTQ Orbitrap Velos (Thermo Fisher Scientific), which was set in data-dependent acquisition mode to perform MS/MS on the top 12 most intense multiply charged cations. RAW data were searched using Sequest on a Sorcerer platform against the Uniprot database. Data were validated and visualized using Byonic software.

## Plasmid construction

To overexpress the protein in *Entamoeba* the full-length coding region of ERM-BP gene (EIN_083100) was cloned into the AvrII and SacII sites in the pEi-CKII-myc plasmid as previously described (*Ehrenkaufer and Singh, 2012*). For gene silencing, the 152-trigger construct was used; full-length coding region of the EIN_083100 gene was cloned downstream of the Trigger region at the NotI and AvrII sites (*Suresh et al., 2016*). Six candidate genes from the LC-MS were cloned in pGEX-4T1 vector at BamHI and NotI sites. The gene IDs and primers used are listed in *Figure 1—source data 2*. Mutants of ERM-BP were generated by site-directed mutagenesis utilizing ERM-BP-pGEX-2T1 as a parent vector. The following substitutions were made at the selected sites by changing to Alanine (A) as D12A, C198A and K150A by using QuickChange Lightning site-directed mutagenesis Kit (Agilent Technologies, Catalog # 210518) and following the manufacturer protocol. An ERM-BP-DBM was generated by changing all five predicted amino acids shown in superscript ($S_AR_L$TKR) into alanine using the Gibson assembly method. Briefly, two PCR reactions were performed by splitting the gene ERM-BP into two fragments, 'A' and 'B'; both share the changed nucleotides for the five amino acids. Fragment-A has 44-nt matching with the adjacent Fragment-B and 5' ends of 'A' and 3' ends of 'B' share 30-nt overlap sequences with the two ends of liner vector pGEX-2T1 (Fragment C). All three fragments (A, B and C) were assembled by using Gibson assembly master mix (New England Biolab, Catalog# E2611S) as manufacturer protocol and transformed into *E. coli*. The construct was confirmed by sequencing and primers are listed in *Figure 1—source data 2*.

## RNA extraction and RT-PCR

Total RNA was extracted from trophozoites and cysts using TRIzol method (Life Technologies). RNA was subjected to DNase treatment (DNase kit; Invitrogen) and reverse transcribed using oligo (dT) primers (Invitrogen). The resultant cDNA (3 µl) was used in subsequent PCRs (25 µl total volume). The number of PCR cycles was set to 30, and 10 µl of PCR products was run on a 1.5% agarose gel. The negative control (minus reverse transcriptase [RT]) was split away before the addition of Superscript RT (Invitrogen) and otherwise treated like the other samples. The primers used in RT-PCR are listed in *Figure 1—source data 2*.

## Expression and purification of recombinant proteins

GST-fusion proteins were expressed in *Escherichia coli* (BL21) and purified by using glutathione beads (GE-healthcare). Briefly, cells were cultured in 100 ml culture media and induced with IPTG (0.5 mM) for 4 hr at 37°C and cells were pelleted. Suspended in 6 ml GST binding buffer [25 mM Tris pH 7.5, 150 mM NaCl, 10 mM MgCl$_2$, 5 mM DTT, 1 mM PMSF, 1X protease inhibitor cocktail (Sigma)] and lysed by sonication. Cell debris was removed by centrifugation at 10,000 rpm for 10 mins at 4°C and the supernatant incubated with 50 µl pre-equilibrated GST-beads. The beads were washed with high-salt solution [500 mM NaCl, 50 mM Tris–HCl (pH7.5), 100 mM PMSF]. Both GST and GST fusion proteins bound to beads were eluted with 500 µl elution buffer [10 mM reduced glutathione, 50 mM Tris-HCl pH 8.0, 10 mM MgCl$_2$, 1 mM PMSF, 1X protease inhibitor cocktail (Sigma)]. Protein was dialyzed over night at 4°C in dialysis buffer [5 mM Hepes pH 7.6, 1 mM DTT, 0.2 mM PMSF, 1 mM EDTA, 10% glycerol] with two changes to remove Glutathione. The protein was quantified and checked by SDS-PAGE and used in all biochemical experiments as mentioned.

## Immunostaining

Myc-tagged ERM-BP expressing trophozoites and cysts were fixed with acetone: methanol (1:1) and permeabilized with 0.1% triton-X100. Cells were incubated with 3% BSA for blocking followed by α-myc antibody from mouse (1:1000) (Cell Signaling) and Alexa 488 α-mouse (1:2500) (Molecular Probes). Localization of the cyst wall Jacob-lectin was carried out by hybridization with the Jacob antibody (at 1:200, a kind gift of John Samuelson), followed by Alexa Fluor 488-conjugated anti-rabbit secondary antibody (1:2500; Molecular Probes). Slides were prepared using Vectashield mounting medium with DAPI (Vector Laboratories, Inc) and visualized using a Leica CTR6000 microscope, using a BD CARVII confocal unit. Images were analyzed using Leica LAS-AF software.

## Measurement of intracellular $NAD^+$/NADH

Intracellular $NAD^+$ and NADH were determined as per the manufacturer's protocol ($NAD^+$/NADH Assay Kit, Cat No: ab65348, Abcam). Briefly $2 \times 10^6$ cells were lysed in 400 µl of $NAD^+$/NADH extraction buffer by sonication (five pulses at 15 amp for 15 s). The lysate was centrifuged at 14,000 rpm and the supernatant containing $NAD^+$/NADH was filtered through a 10kD spin column to get rid of enzymes, which may consume NADH rapidly. To detect the NADH in the sample a decomposition step was performed by heating the samples at 60°C for 30 min; under this condition, all the $NAD^+$ will be decomposed while NADH will be still intact. 100 µl of reaction mix was prepared for each standard and samples in duplicates in a clear bottom 96-well plate. The plate was incubated at room temperature for 5 min to convert $NAD^+$ to NADH followed by addition of 10 µl NADH developer into each well and incubated at room temperature for 2 hr. OD was measured at 450 nm using a plate reader (BioTek Cytation3). Concentration of $NAD^+$t and NADH in the test samples is calculated from standard curve and actual concentration was determined as: $NAD^+$t concentration = ($NAD^+$t/$Sv$)*D; NADH concentration = (NADH/$sv$)*D; $NAD^+$ = $NAD^+$t - NADH; $NAD^+$/NADH ratio = ($NAD^+$t-NADH)/NADH. Where: $NAD^+$t = total amount of $NAD^+$ in the sample well calculated from standard curve (pmol); NADH = amount of NADH in the sample well calculated from standard curve (pmol); Sv = sample vol added to the reaction well (µl); D = sample dilution factor.

## Enzymatic assay

Enzymatic assay using ERM-BP was performed as described earlier (*Zhang et al., 2008*). Briefly, the reaction was performed in 30 mM Tris-HCl buffer (pH 7.5). For each reaction 160 µg recombinant protein and 20 mM nicotinamide was used as substrate, incubated at 37°C for 1 hr and reaction terminated by adding 20 µl of trichloroacetic acid (80%, v/v). The protein was precipitated by centrifugation at 13,000 RPM for 10 min followed by filtration of the pellet. Samples were diluted 1:25 in 50% acetonitrile/water and 25 µl were injected into an Agilent 1100 Series HPLC system (Agilent Technologies) and separated on a Luna 3 µm C18(2) 100 Å column with a 5 – 95% gradient of acetonitrile in water with 0.1% trifluoroacetic acid, 7.5 min at a flow rate of 200 µl/min and peaks detected by UV light at 254 nm. Area under the curve values for the nicotinamide peaks were calculated for each condition using Analyst software (SCIEX) and normalized by dividing obtained values by those obtained under negative control conditions (GST). Percentage substrate conversion values were calculated by subtracting normalized AUC values from 1. Any obtained negative % conversion values resulting from baseline substrate AUC variability are shown as 'no conversion'. For initial confirmation of substrate and product peaks LC-analysis was coupled to MS-analysis on an API 150EX single-quadrupole mass spectrometer (Applied Biosystems) with an electrospray interface.

## Protein thermal stability shift assay

Protein thermal stability shift assay was performed as per the manufacturer's protocols (Protein thermal Shift starter kit, Cat no: 4462263, Applied Biosystems). In brief, all the reactions were set up in final volumes of 20 µl in 96-well plates. In each well, 1 µg of purified recombinant protein was added with 5 µl of thermal stability buffer and 2.5 µl of 8X dye with varying amounts of $NAD^+$ (0, 1, 2 and 4 mM). For each set, samples were made in quadruple and at least three biologically independent experiments were conducted. Temperature gradient was set in the range of 25°C to 99°C with a 0.05°C/s ramp rate using real-time PCR system (Applied Biosystems). Melting temperature (Tm) was calculated by using Protein Thermal Shift Software 1x (Applied Biosystems).

## Acknowledgements

The authors are grateful to Chris Adams and Ryan Leib, Stanford University Mass Spectrometry facility for their help with Mass spectrometry. The authors also thank John Samuelson, Boston University for the kind gift of *Entamoeba* Jacob antibodies. We thank Matthew Bogyo, Stanford University for access to the HPLC equipment and the entire Singh lab for critical reading of the manuscript.

## Additional information

### Funding

| Funder | Grant reference number | Author |
| --- | --- | --- |
| National Institute of Allergy and Infectious Diseases | R21AI119893 | Upinder Singh |
| National Institute of Allergy and Infectious Diseases | R21AI117171 | Upinder Singh |

The funders had no role in study design, data collection and interpretation, or the decision to submit the work for publication.

### Author contributions

Dipak Manna, Conceptualization, Formal analysis, Validation, Investigation, Methodology, Writing—original draft, Writing—review and editing; Christian Stephan Lentz, Conceptualization, Investigation, Methodology, Writing—review and editing; Gretchen Marie Ehrenkaufer, Conceptualization, Formal analysis, Investigation, Methodology, Writing—review and editing; Susmitha Suresh, Amrita Bhat, Investigation, Methodology; Upinder Singh, Conceptualization, Resources, Supervision, Funding acquisition, Investigation, Project administration, Writing—review and editing

### Author ORCIDs

Dipak Manna http://orcid.org/0000-0001-5112-3957
Upinder Singh http://orcid.org/0000-0003-0630-0306

### Decision letter and Author response

Decision letter https://doi.org/10.7554/eLife.37912.029
Author response https://doi.org/10.7554/eLife.37912.030

## Additional files

### Supplementary files

• Transparent reporting form
DOI: https://doi.org/10.7554/eLife.37912.027

All data generated or analysed during this study are included in the manuscript and supporting files. Source data files have been provided for all data discussed in the manuscript.

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
