## [Decision Letter]

Thank you for submitting your article "An NAD+dependent novel transcription factor controls stage conversion in Entamoeba" for consideration by *eLife*. Your article has been reviewed by three peer reviewers, one of whom is a member of our Board of Reviewing Editors and the evaluation has been overseen by Gisela Storz as the Senior Editor. The following individual involved in review of your submission has agreed to reveal her identity: Christine Clayton (Reviewer #1).

The reviewers have discussed the reviews with one another and the Reviewing Editor has drafted this decision to help you prepare a revised submission. The first part is the summary from the reviewers; the remainder consists of detailed comments.

Summary:

Encystation is a cellular differentiation process from the proliferative, motile, and disease-causing stage to the dormant stage. Despite the importance of the process, the molecular events that occur during encystation have not been fully elucidated at the molecular levels. Previously Coppi et al., (2002) demonstrated several lines of evidence that an autocrine catecholamine system is involved in Entamoeba encystation. Mi-ichi et al., (2015) reported that cholesteryl sulfate plays an important role in Entamoeba encystation. In this study Manna et al. utilized RNA-Seq data to identify the molecular triggers involved in regulating stage conversion in Entamoeba. They identify a transcription factor, Encystation Regulatory Motif-Binding Protein (ERM-BP), that regulates encystation. This is an important observation with respect to parasite biology. In general, the work is experimentally solid and clearly presented.

The authors report the identification of Encystation Regulatory Motif-Binding Protein as a transcription factor that regulates encystation, the transition of trophozoites of Entameba histolytica to the dormant cyst form. This developmental transition is key to the life cycle and transmission of ameba as trophozoites are pathogenic but cysts are transmitted. Very little is known about the regulators of developmental transitions in pathogenic protozoa and nearly nothing is known about this transition in this important parasite that causes amebiasis.

The authors use a combination of complementary techniques to identify ERM-BP. They identified 9 motifs enriched upstream of induced transcripts and tested these in EMSA. Of these only one showed specific binding to nuclear extracts as demonstrated in a series of carefully controlled experiments. The motif was validated in reporter assays and then used to identify the binding protein by MS. The importance of the TF was tested using gene silencing and confirmed lower cyst numbers and cysts that were less robust. They go on to show that the binding of NAD+ to ERM-BP alters the conformation and facilitates binding to promoter DNA. Then they follow up with ERM-BP mutant studies that map DNA binding and nicotinomidase activity.

To link the biochemical properties of ERM-BP to the biology of the parasite, they show that cellular NAD+ increases during encystation and that exogenous NAD+ promotes encystation. Finally, ERM-BP catalyzes nicotinamide to nicotinic acid implicating these metabolites in potential signaling during stage differentiation. Altogether this paper does a nice job providing mechanistic insights into Entameba encystation.

Essential revisions:

Reviewer #1:

Annotated PDF review of paper is attached. The comments mostly suggest minor modifications.

Reviewer #2:

The authors reported that they chose early data (24hours) during encystation but according to previous publication by other groups the early period of encystation is 0.5 to 8hours when the trophozoites form large multicellular aggregates. In their previous publication the authors also have 8hours, 48hours and 72hours RNA seq data during encystation. What is the reason behind choosing the 24hours time point if early time points are available?

Subsection “Identification of ERM-binding protein” "EIN_083100 is a hypothetical.………..during encystation" The authors cited two references De Cadiz et al., 2013 and Ehrenkaufer et al., 2013. De Cadiz et al., didn't mentioned about this gene and also Ehrenkaufer et al., 2013 showed that this gene mostly up regulate at 72hours not 24hours during encystation.

Regarding Figure 3C and Figure 4A instead of showing relative percentage of encystation the authors should show actual percentage of encystation.

The authors completely silenced ERM-BP (Figure 4—figure supplement 1) and reported that loss of ERM-BP significantly decreased the encystation efficiency. If this gene is important for encystation, why didn't knock down of this gene didn't completely shut down the encystation process.

Reviewing editor comment: unless both genes were deleted, there is probably some of the gene product left – even if it isn't detected using the methods used.

The authors further reported that the defect is in cyst wall formation. Do all the cysts contain the defective cyst wall?

The authors reported that silencing of ERM-BP decreased parasite viability in encystation media however they also reported that it also decreased the encystation efficiency. In my opinion the decreased in encystation efficiency is due to growth defect caused by silencing of ERM-BP.

Subsection “Intracellular NAD+/NADH is elevated during encystation and NAD+ facilitates Encystation”: The cited Ref. Jeelani and Nozaki demonstrates in contradiction to what is stated here that NAD/NADH ratio is significantly elevated during encystation. Cite the original ref. Jeelani et al., 2012. However, in that study only NAD and NADP concentration are reported and the NAD concentration is little bit increases from 0 to 2hours (2.296 to 2.978 nmoles) and then it started decreasing 8hours (1.669 nmoles), 24 (0.614nmoles), 48 (0.497 nmoles) and 120h (0.243 nmoles).

In Figure 5A, the authors mentioned NAD/NADH ratio. What about the actual NAD and NADH concentration in the trophozoites and the cyst? Why the authors used 1mM extracellular NAD to the encysting parasites (Figure 5B).

The authors showed that ERM-BP produce nicotinic acid from nicotinamide and propose that nicotinic acid may also have important roles in encystation. However, the authors didn't show any work how nicotinic acid may act a second messenger and is involved in encystation. In my opinion Nicotinc acid produced by ERM-BP may convert to NAD using NAD biosynthesis pathway. May be expression of all the enzymes in this pathway is affected upon ERM-BP silencing. Therefore, it will be interesting to perform RNA seq and may be metabolome analysis of the ERM-BP_OX and silenced cell lines.

Subsection “Identification of ERM-binding protein” The authors mention "source data 6" but the source data 6 contain summary of proteins identified by LC-MS and confirmation.

Subsection “Domain characterization, NAD+/DNA binding and catalytic activity of ERM-BP”, the sentence “We also observed that the NAD+ and ERM-BP binding is very specific as NADP did not result in a protein thermal shift in the Tm” NADP or NAD?

*Reviewer #3:*

1) What can they tell us about the genes that contain the ERM-BP motif? Are there specific pathways upregulated?

2) What can they tell us about proteins ID in mass spec? No annotations are provided. Are there proteins expected to be important for gene regulation? Criteria used to ID proteins are not very stringent (one peptide at 95% confidence) so they should provide more detail about the methodology for peptide ID and provide more MS data (number of peptides, confidence, etc.) instead of only providing fold change.

3) Figure 1. Generally, the experiments are well performed and controlled, but it is surprising that Mut2 seems to be a better competitor than the WT motif. Do they have an explanation for this? They could explore this with additional mutant experiments/luciferase studies, or search for that motif in cyst genes?

Generally, the experiments are rigorously performed and statistical analysis is provided.

---

## [Author Response]

Essential revisions:

Reviewer #2:

The authors reported that they chose early data (24hours) during encystation but according to previous publication by other groups the early period of encystation is 0.5 to 8hours when the trophozoites form large multicellular aggregates. In their previous publication the authors also have 8hours, 48hours and 72hours RNA seq data during encystation. What is the reason behind choosing the 24hours time point if early time points are available?

We appreciate that this is an important point to clarify. The reason behind choosing the 24hours time points was to avoid the general stress response (i.e. osmotic stress) that may be a confounding factor in the earlier time points (i.e. 2hours and 8hours). So, as a first attempt at proving that our approach (identifying transcriptional networks based on motif and protein identification) could work we picked the first time point at which morphologically distinct cysts are seen in substantial number. In this way, we hoped to enrich the transcriptional control mechanisms for those that regulate cyst formation (rather than general stress response). We did not analyze later time points (i.e. 48hours or 72hours) as lysing these cysts is more difficult and can hamper nuclear extract generation. However, now that our approach and workflow has been validated, earlier or later time points can certainly be explored in the future.

Subsection “Identification of ERM-binding protein” "EIN_083100 is a hypothetical.………..during encystation" The authors cited two references De Cadiz et al., 2013 and Ehrenkaufer et al., 2013. De Cadiz et al., didn't mentioned about this gene and also Ehrenkaufer et al., 2013 showed that this gene mostly up regulate at 72hours not 24hours during encystation.

The gene ID EIN_083100 was previously named EIN_052150 (AmoebaDB). We have indicated the previous EIN nomenclature when we first mention the gene in our manuscript (subsection “Identification of ERM-binding protein”). The RNA expression levels are listed in the supplementary tables in of De Cadiz et al., 2013 and Ehrenkaufer et al., 2013. In the Ehrenkaufer manuscript, this gene was largely regulated at later time point of encystation (i.e. 72hours). However, our RT-PCR results show that EIN_083100 RNA expression is undetectable in trophozoites and up-regulated as early as 24hours of encystation (Figure 4—figure supplement 2A). Whether this difference is due to technical differences in RNA-Seq versus RT-PCR or changes in parasites that have occurred over several years is not clear. However, the overall point is that this gene is developmentally regulated.

Regarding Figure 3C and Figure 4A instead of showing relative percentage of encystation the authors should show actual percentage of encystation.

We appreciate this concern. In our manuscript, the encystation efficiency was determined by automated quantitative imaging in 96-well plates as described in earlier studies (Ehrenkaufer et al., 2018; Suresh et al., 2016). In this approach, encystation efficiency or cysts percentage compared to total cell count (i.e. those that remained as trophozoites) cannot be calculated. However, we do have the data for the actual number of cysts for each experiment and have changed the figures to show the actual cyst numbers. These changes have been made in Figure 3C, Figure 4A and Figure 5C. In the revised figures the actual number of cysts are presented from three biological experiments with eight replicates for each biological experiment (subsection “Parasite culture, transfection and induction of stage conversion”).

The authors completely silenced ERM-BP (Figure 4—figure supplement 1) and reported that loss of ERM-BP significantly decreased the encystation efficiency. If this gene is important for encystation, why didn't knock down of this gene didn't completely shut down the encystation process.Reviewing editor comment: unless both genes were deleted, there is probably some of the gene product left – even if it isn't detected using the methods used.

We agree with the reviewing editor comment. We also point out that the ERM-BP is likely not the only gene that regulates encystation and so some level of cyst formation can still happen.

The authors further reported that the defect is in cyst wall formation. Do all the cysts contain the defective cyst wall?

We appreciate the comment. In order to get quantitative data on the percentage of cysts with defective cyst wall, we did quantitative analysis by staining the cyst wall using two markers: one with the calcofluor white that stains the chitin in the cyst wall and a second with an antibody to the cyst wall protein, Jacob. Using these two markers, we assessed the number of morphologically normal and defective cysts in the control and ERM-BP downregulated cell lines. The data are shown in Figure 4D.

The authors reported that silencing of ERM-BP decreased parasite viability in encystation media however they also reported that it also decreased the encystation efficiency. In my opinion the decreased in encystation efficiency is due to growth defect caused by silencing of ERM-BP.

We appreciate the opportunity to clarify this issue. Parasites with ERM-BP silenced have no obvious growth defects under trophozoite conditions. This suggests that trophozoites, which are starting to encyst but getting arrested in development have reduced viability. This point has been clarified in the text (subsection “Silencing of ERM-BP expression levels reduces encystation efficiency and produces ghost like cysts”). We also show that the cysts produced in ERM-BP downregulated parasites are defective in excystation. Thus, overall, the downregulation of ERM-BP results in fewer cysts and even the cysts that are produced are largely defective in viability and ability to excyst.

Subsection “Intracellular NAD+/NADH is elevated during encystation and NAD+ facilitates Encystation”: The cited Ref. Jeelani and Nozaki demonstrates in contradiction to what is stated here that NAD/NADH ratio is significantly elevated during encystation. Cite the original ref. Jeelani et al., 2012. However, in that study only NAD and NADP concentration are reported and the NAD concentration is little bit increases from 0 to 2hours (2.296 to 2.978 nmoles) and then it started decreasing 8hours (1.669 nmoles), 24 (0.614nmoles), 48 (0.497 nmoles) and 120h (0.243 nmoles).

Thanks to the reviewer for pointing to this issue. We have included the original ref. Jeelani et al., 2012 in revised manuscript. In this paper Jeelani et al., reported slight increase of NAD^+^ from 0 to 2hours and in latter time points of encystation the NAD^+^ level decreased gradually. It is true that our findings do not match those of the previous publication, but there are some clarifications that need to be made:

(i) Importantly, we used two different assay methods: NAD^+^/NADH assay kit (Abcam, Cat no: ab65348) and NAD^+^/NADH-Glo assay (Promega, Cat no: G9071). Both gave us consistent results.

(ii) Furthermore, we have now also measured the NAD^+^ and NADH concentration in sarkosyl resistant 24 h cysts and see the similar elevated amount of NAD^+^ in the cysts, supporting our initial results as reported in the paper (now added to Figure 5—figure supplement 1).

(iii) Overall, our results using our parasites, encystation efficiency etc are all highly internally consistent: NAD^+^ levels increase during encystation, increasing NAD^+^ increases ERM-BP binding to DNA, and increasing NAD^+^ increases cyst formation.

(iv) There are some possible explanations of the apparent disparity of our results with Jeelnai et al., 2012.

a) First, the methodology used by Jeelani et al., 2012 to identify metabolomics used 75% chilled methanol to quench the metabolic activity. This also fixes cells and NAD^+^ may leach out during this fixation. Thus, in the latter step when the metabolites are extracted with chloroform and water by sonication to lyse the cyst, the results may not accurately measure NAD^+^ levels.

b) Second, Jeelani et al., 2012 didn’t mentioned the concentration of NADH. NAD^+^ is very unstable and can be easily reduced to NADH and this may lead to an apparent artificial reduction of NAD^+^.

In Figure 5A, the authors mentioned NAD/NADH ratio. What about the actual NAD and NADH concentration in the trophozoites and the cyst?

The actual concentration of NAD^+^ and NADH are now mentioned in the manuscript (concentrations of NAD^+^ and NADH in trophozoites are 1.5 ± 0.34 nmole and 2.3 ± 0.75 nmole respectively and in 24-hour cysts, the NAD^+^ and NADH concentrations are 4.1 ± 1.40 nmole and 1.2 ± 0.40 nmole respectively per 2X10^6^ cells). We have now mentioned the actual concentrations of NAD^+^ and NADH in the manuscript (subsection “Intracellular NAD+/NADH is elevated during encystation and NAD+ facilitates encystation”).

Why the authors used 1mM extracellular NAD to the encysting parasites (Figure 5B).

This amount of NAD^+^ is often used in experiments to determine the effect of exogenous NAD (Bilington et al., 2008; Ying et al., 2003). Additionally, there are data that extracellular NAD^+^ can enter into cells and result in millimolar intracellular concentrations (Ying et al., 2003). Additionally, we have measured the concentration of intracellular NAD^+^ in *Entamoeba* cells treated with 1 mM NAD^+^. Our result shows a gradual increase of intracellular NAD^+^ up to 8hours (New Figure 5B).

The authors showed that ERM-BP produce nicotinic acid from nicotinamide and propose that nicotinic acid may also have important roles in encystation. However, the authors didn't show any work how nicotinic acid may act a second messenger and is involved in encystation. In my opinion Nicotinc acid produced by ERM-BP may convert to NAD using NAD biosynthesis pathway. May be expression of all the enzymes in this pathway is affected upon ERM-BP silencing. Therefore, it will be interesting to perform RNA seq and may be metabolome analysis of the ERM-BP_OX and silenced cell lines.

The reviewer raises a number of valuable points. Thank you. We agree that there is much work to be done including the RNA-Seq experiments that the reviewer suggests, although we feel that these are beyond the scope of this manuscript. We have modified the model to indicate that nicotinic acid can also convert to NAD^+^. We agree that we have not done the definitive experiments to demonstrate that nicotinic acid has an important role in enystation, and have thus indicated with a ? in the model. This and many other follow-up experiments are undoubtedly needed and will form the basis of the next manuscript. Thank you for the comments and suggestions.

Subsection “Identification of ERM-binding protein” The authors mention "source data 6" but the source data 6 contain summary of proteins identified by LC-MS and confirmation.

Thanks to the reviewer for this correction. This has been removed.

Subsection “Domain characterization, NAD+/DNA binding and catalytic activity of ERM-BP”, the sentence “We also observed that the NAD+ and ERM-BP binding is very specific as NADP did not result in a protein thermal shift in the Tm” NADP or NAD?

We clarify that this is NADP^+^. Thermal stability assays with ERM-BP and NADP^+^ are included in Figure 6—figure supplement 2G.

Reviewer #3:

1) What can they tell us about the genes that contain the ERM-BP motif? Are there specific pathways upregulated?

Yes, this is an important point to clarify. The genes that contain ERM in their promoters are listed as Figure 1—source data 3 with annotation and molecular weight. We have identified 131 genes having ERM motif in their promoter. Out of 131 genes 65 (~50%) are annotated as hypothetical. Of the remainder, some genes are annotated as cyst wall proteins (EIN_066080: Chitin binding lectin, EIN_284810: Chitin synthase), and we do see a reduction in RNA level by RT-PCR (Figure 4—figure supplement 2A). Additionally, a few genes are associated with metabolism and stress response (Figure 1—Source data 3). However, considering 50% as hypothetical genes it is difficult to speculate whether any specific pathways upregulated. Future studies with RNA-Seq of ERM-BP downregulated parasites will give us a better idea about the transcriptional network associated with the ERM-BP transcription factor.

2) What can they tell us about proteins ID in mass spec? No annotations are provided. Are there proteins expected to be important for gene regulation? Criteria used to ID proteins are not very stringent (one peptide at 95% confidence) so they should provide more detail about the methodology for peptide ID and provide more MS data (number of peptides, confidence, etc) instead of only providing fold change.

We have included the annotation, peptide number in the Figure 2—source data 1 as the reviewer suggested. However, it is important to note that the mass-spectrometry was performed from the gel-shifted band with the idea that this was a starting point to identify candidate proteins. The criteria we used are relatively standard, however, the point is less about the stringency of our initial mass spec analysis but that this allowed us to identify proteins that could potentially bind to ERM-BP. Only six proteins consistently showed up in all three biological experiments and were considered candidate proteins for binding to ERM (Table 1). Subsequently, we have done extensive experiments to prove that EIN_083100, does in fact bind ERM and has important biological effects.

3) Figure 1. Generally, the experiments are well performed and controlled, but it is surprising that Mut2 seems to be a better competitor than the WT motif. Do they have an explanation for this? They could explore this with additional mutant experiments/luciferase studies, or search for that motif in cyst genes?Generally, the experiments are rigorously performed and statistical analysis is provided.

Thank you for the positive comments about the paper. We agree with the reviewer that Mut-2 looks as a better competitor at 10X excess probe in this particular gel. However, it is important to note that at 100X excess probe it doesn’t show much difference compared to WT competitor. Importantly, we have noted in multiple other gel shift assays that Mut-2 acts similar to WT-motif as a competitor (data not shown). Thus, the apparent difference in this gel may be due to unequal loading of nuclear extract or probe. This point has been clarified in the manuscript (subsection “Motif-2 specifically binds to cyst nuclear protein(s)”).